# From Kolmogorov to Cauchy: Shallow XNet Surpasses KANs

**Xin Li**[*]
College of Computer Science and Technology
Dongguan University of Technology, China
Institute for Advanced Research
Great Bay University, Dongguan, Guangdong, China
xinli2023@u.northwestern.edu

**Xiaotao Zheng**[*]
Center for Financial Engineering
Soochow University, Suzhou, China
20234013002@stu.suda.edu.cn

**Zhihong Xia**[†]
Institute for Advanced Research
Great Bay University, Dongguan, Guangdong, China
Department of Mathematics
Northwestern University, Evanston, IL, USA
xia@math.northwestern.edu

## Abstract

We study a shallow variant of XNet, a neural architecture whose activation functions are derived from the Cauchy integral formula. While prior work focused on deep variants, we show that even a single-layer XNet exhibits near-exponential approximation rates—exceeding the polynomial bounds of MLPs and spline-based networks such as Kolmogorov–Arnold Networks (KANs).

Empirically, XNet reduces approximation error by over 600× on discontinuous functions, achieves up to 20,000× lower residuals in physics-informed PDEs, and improves policy accuracy and sample efficiency in PPO-based reinforcement learning—while maintaining comparable or better computational efficiency than KAN baselines.

These results demonstrate that expressive approximation can stem from principled activation design rather than depth alone, offering a compact, theoretically grounded alternative for function approximation, scientific computing, and control.

## 1   Introduction

As deep learning models grow increasingly large and resource-intensive, there is renewed interest in shallow architectures that combine theoretical rigor with practical efficiency. In particular, scientific applications such as PDE solving, symbolic regression, and control require models that are not only expressive but also interpretable, sample-efficient, and robust to irregularities.

Traditional universal approximators like shallow MLPs [5, 10] suffer from slow convergence [1, 7], especially in high dimensions. Recent efforts such as Kolmogorov–Arnold Networks (KANs) [19] improve expressivity via spline-based activations, but still rely on multi-layer composition and struggle with discontinuities due to their smooth basis structure.

---

[*]Equal contribution
[†]Corresponding author

In this work, we study a principled *single-layer* variant of **XNet**, a neural architecture built on the *Cauchy activation function* introduced in [16]:

$$\phi(x) = \frac{\lambda_1 x + \lambda_2}{x^2 + d^2}, \quad \lambda_1, \lambda_2, d \in \mathbb{R}.$$

While [16] proposed Cauchy activations as a plug-and-play component for general deep architectures, we focus on the *shallow, single-layer* setting and provide the first theoretical analysis of its approximation rates. We prove that even a single-layer XNet achieves arbitrarily high-order convergence $O(N^{-r})$ for any $r > 0$ on real-analytic functions, and demonstrate its effectiveness across function approximation, PDE solving, and reinforcement learning tasks.

These rational functions—derived from the Cauchy integral formula—yield smooth, localized, and trainable basis elements, enabling XNet to approximate irregular, high-dimensional, and even discontinuous functions with far fewer neurons and no depth.

**Main theoretical result.** We prove that XNet achieves *arbitrarily fast* convergence $O(N^{-p})$ for any $p > 0$ on smooth target functions. In contrast, classical ReLU networks are limited by the curse of dimensionality with rate $O(N^{-2r/d})$ (for target smoothness $r$ and input dimension $d$), and spline-based KANs achieve $O(N^{-k})$ constrained by the fixed spline degree $k$. XNet's analytic Cauchy basis functions overcome both limitations by enabling tunable, high-order approximation in a single layer. See Section 3.2 for the formal result and Appendix A.5 for the proof.

Figure 1 illustrates this convergence behavior: as the number of neurons $N$ increases, XNet maintains a significantly steeper error decay rate compared to ReLU and KAN.

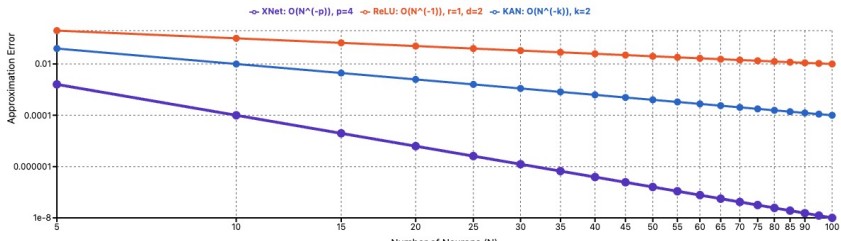

Figure 1: Comparison of approximation rates. XNet achieves $O(N^{-p})$ with $p = 4$, outperforming ReLU ($O(N^{-2r/d})$ with $r = 1$, $d = 2$) and KAN ($O(N^{-k})$ with $k = 2$). Log-log plot shows approximation error versus neuron count $N$.

**Empirical results:** Across three domains, XNet outperforms strong baselines:

- **Function approximation:** over **600×** lower MSE on discontinuous targets.
- **PDE solving:** over **10,000×** lower error than deep MLPs in PINN settings. (see Appendix A.1 for computational analysis)
- **Reinforcement learning:** faster convergence and higher rewards under PPO.

These findings challenge the depth-centric paradigm, showing that *activation design alone* can enable expressive, compact, and theoretically grounded models—often with computational efficiency comparable to or better than multi-layer alternatives.

**Why this matters now.** As foundation models encounter scaling bottlenecks, there is renewed interest in shallow networks that combine interpretability, efficiency, and provable expressivity. XNet exemplifies this direction by achieving strong performance without depth or overparameterization.

[16] introduced the Cauchy activation and established its universal approximation properties. Building on this foundation, we establish new convergence rates in dual function spaces (Theorem 3.2; proof in Appendix A.5), proving $O(N^{-r})$ for any $r > 0$ on real-analytic functions—the first such result for shallow XNet. While our analysis focuses on single-layer networks, [16] demonstrated the modularity of Cauchy activations in deeper architectures (multi-layer MLPs, CNNs, ResNets) on MNIST and CIFAR-10.

Together, these results challenge the depth-centric paradigm, showing that principled activation design alone can yield compact, expressive models for scientific and symbolic AI.

## 2 Related Work

**Universal Approximation and Expressive Efficiency.** Shallow MLPs with non-polynomial activations are universal approximators [5, 10], but their approximation efficiency degrades in high dimensions [1, 26]. Recent work improves expressivity through spline [19], rational [3], and interpolant-based [23] activations. Parametric activations [2, 15] offer adaptability, but often increase complexity or lack convergence guarantees. XNet addresses this gap by connecting activation design with provable, high-order approximation in a shallow architecture.

**Structured Activation Networks.** KANs [19] approximate functions via learnable B-splines along edges, enabling efficient approximation with compositional depth. However, their reliance on spline smoothness and fixed grid resolution limits adaptability to discontinuities or noise. Recent applications span time-series [25], RL [11], and generative modeling [9]. In contrast, XNet provides analytic, localized responses with no depth or grid assumptions.

**Rational and Complex-Valued Models.** Rational activations [4, 3] have been used to improve approximation power via learnable Padé-like functions. However, these approaches are primarily heuristic and often lack theoretical guarantees beyond empirical smooth function fitting. Our work draws from complex analysis—specifically the Cauchy integral formula—to derive rational activations with localized and tunable responses. This analytic foundation enables high-order approximation in both classical and dual function spaces, distinguishing XNet from prior rational models.

**Scientific Machine Learning and Operator Learning.** PINNs [22] and neural operators [17, 14] enable learning in scientific domains, but often rely on deep networks, spectral bias, or task-specific assumptions. Spectral solvers [8] and low-rank architectures [27] improve convergence, but require expert tuning. XNet offers a shallow alternative with strong theoretical grounding and minimal tuning—achieving fast convergence across physics-informed and control tasks.

## 3 Theoretical Foundations of XNet

### 3.1 Limitations of Spline-Based KANs

Kolmogorov–Arnold Networks (KANs) [19] implement the classical Kolmogorov superposition theorem [13] via spline-based edge activations:

$$f(x) = \sum_q \Phi_q \left( \sum_p \phi_{q,p}(x_p) \right), \quad \phi(x) = w_b b(x) + w_s \sum_i c_i B_i(x),$$

where $B_i(x)$ are B-spline basis functions. While KANs offer adaptive local representations and $O(N^{-k})$ convergence for spline degree $k$, they suffer from poor generalization near discontinuities, require multi-layer structures, and incur high computational costs due to spline interpolation and grid dependencies.

### 3.2 Cauchy Basis Architecture (XNet)

This section analyzes a shallow variant of **XNet**, a neural architecture built on *Cauchy basis activations* derived from the Cauchy integral formula:

$$\phi(x) = \frac{\lambda_1 x + \lambda_2}{x^2 + d^2}, \quad \text{with trainable } \lambda_1, \lambda_2, d.$$

This yields localized, tunable, and analytically smooth responses. Below, we present the theoretical foundations supporting its expressivity.

**(1) Approximation Efficiency.** XNet achieves arbitrarily fast convergence $O(N^{-p})$ on smooth target functions, where the convergence exponent $p > 0$ depends on the analyticity (i.e., smoothness) of the target. This surpasses the $O(N^{-2r/d})$ rate of ReLU MLPs [1, 26] and the $O(N^{-k})$ rate of spline-based KANs [19], where $r$ is the Sobolev smoothness and $d$ is the input dimension.

See Appendix A.2 for a detailed derivation.

**(2) Localization and Noise Adaptivity.** Cauchy activations exhibit rational decay $O(1/x)$, yielding naturally localized basis functions:

$$\phi_i(x) = \frac{\lambda_{1,i}(x - \mu_i) + \lambda_{2,i}}{(x - \mu_i)^2 + d_i^2}.$$

By adjusting the parameter $d_i$, neurons can adapt their receptive field: small $d_i$ captures sharp discontinuities, while large $d_i$ smooths over noisy regions. Unlike B-splines, which require predefined grid partitions, XNet offers neuron-wise adaptivity for heterogeneous signals.

See Appendix A.3 for analytical expressions and derivative properties.

**(3) Comparison with B-Spline KANs.** We contrast the analytical and computational properties of XNet (Cauchy basis) and Kolmogorov–Arnold Networks (B-spline basis) in Table 1.

Table 1: Comparison of XNet vs. B-spline KANs

| Aspect | Cauchy Basis (XNet) | B-spline (KAN) |
|---|---|---|
| Approximation Rate | $O(N^{-p})$, arbitrary $p > 0$ | $O(N^{-k})$, limited by degree $k$ |
| Basis Adaptivity | Continuous, localized, trainable | Grid-constrained, piecewise |
| Matrix Structure | Dense (condition number $O(N)$) | Sparse (condition number $O(1)$) |
| Derivative Computation | Closed-form, differentiable w.r.t. $d^2$ | Piecewise, non-smooth |
| Architectural Cost | Single-layer, low parameter count | Multi-layer, higher cost |

This comparison highlights XNet's key advantages over KANs: global analytical structure, tunable locality, and full differentiability—without reliance on grid resolution. For further details on matrix conditioning and gradient properties, see Appendix A.4.

**(4) Cauchy Approximation Theorem.**

**Theorem 3.1** (Th.1 in [16]). *Let $f(z^1, \ldots, z^d)$ be analytic on open $U \subset \mathbb{C}^d$. Then for any $\varepsilon > 0$, there exists a finite sum:*

$$\sup_{z \in U} \left| f(z) - \sum_{k=1}^{N} \frac{\lambda_k}{\prod_{j=1}^{d}(\xi_k^j - z^j)} \right| < \varepsilon, \quad \text{with} \quad \varepsilon = O(N^{-p}).$$

This theorem shows that Cauchy activation functions can approximate analytic functions with arbitrarily high order. The specific form

$$\phi(x) = \frac{\lambda_1 x + \lambda_2}{x^2 + d^2}$$

was introduced in [16] as a simplified instantiation.

While the above result applies to partial fractions, it does not directly yield rates for XNet. Our Theorem 3.2 provides the first constructive $O(N^{-r})$ rate for single-layer XNet on real-analytic functions $f \in C^\omega(K)$ over compact domains $K \subset \mathbb{R}^d$.

**(5) New Result: High-Order Approximation of XNet.** We now prove that a one-hidden-layer neural network using Cauchy activations inherits the same approximation rate.

**Theorem 3.2** (High-Order Approximation of XNet). *Let $K \subset \mathbb{R}^d$ be compact, and let $f \in C^\omega(K)$ be real analytic. Let $\Phi = \left\{ \frac{\lambda_1 x + \lambda_2}{x^2 + d^2} \right\}$ be the Cauchy activation family. Then for any $r > 0$, there exists a one-hidden-layer XNet function $f_N \in \text{span}(\Phi)$ such that*

$$\|f - f_N\|_{C^0(K)} = O(N^{-r}).$$

This is a new result: while Theorem 2 in [16] shows that XNet is universal, it does not quantify convergence rate. Our theorem gives an explicit $O(N^{-r})$ bound for real-analytic $f$.

*Sketch.* (1) Use the Cauchy approximation theorem to approximate any 1D analytic function with rate $O(N^{-r})$; (2) Represent multivariate polynomials using $(w^\top x)^k$, reducing to the 1D case; (3) Approximate $f$ by polynomials, then apply Step (2). See Appendix A.5 for details. $\square$

*Remark* 3.3. While our theorem assumes $f \in C^\omega(K)$ (real-analytic), empirical results (Section 4.1.1) show XNet performs well even on discontinuous functions like the Heaviside step. This aligns with classical results on rational approximation to non-smooth functions (e.g., [20, 24] for $|x|^\alpha$), which demonstrate near-exponential rates. Extending our theoretical framework to broader function classes is an important direction for future work.

**(6) Neuron Efficiency at Arbitrary Approximation Orders.** We show that one-hidden-layer XNet achieves convergence rates of $O(N^{-r})$ for any desired order $r > 0$. To achieve an approximation error of $\epsilon$, the number of required neurons satisfies:

$$N = O\left(\epsilon^{-1/r}\right), \quad \forall r > 0.$$

This tunable convergence order distinguishes XNet from classical architectures: ReLU MLPs are fixed at $r = 1/d$ [1], while spline-based KANs are limited by spline degree $k$, i.e., $r \leq k$.

In contrast, XNet leverages the analytic structure of the Cauchy basis to achieve flexible, localized, and high-precision approximation with fewer neurons. This result is particularly beneficial for smooth, high-dimensional, or irregular targets. A complete derivation of these scaling laws is provided in Appendix A.6.

Table 2: Theoretical comparison of approximation rate and neuron complexity.

| Architecture | Approx. Rate | Neurons for Error $\epsilon$ | Adjustable Order $r$? | Depth Requirement |
|---|---|---|---|---|
| ReLU MLP | $O(N^{-1/d})$ | $O(\epsilon^{-d})$ | ✗ (fixed) | Deep |
| KAN (Spline) | $O(N^{-k})$ | $O(\epsilon^{-1/k})$ | ✗ (limited by $k$) | Moderate |
| **XNet (Ours)** | $O(N^{-r})$ | $O(\epsilon^{-1/r})$ | ✓ (Any $r > 0$) | **Single-layer** |

*Remark* 3.4. The ability to *tune the approximation order arbitrarily* via $r$ is a direct consequence of the Cauchy approximation theorem [16], which guarantees exponential convergence for analytic functions.

This theoretical framework not only guarantees approximation efficiency, but also explains XNet's empirical strength in modeling irregular and high-dimensional scientific data.

## 4 Experimental Setup and Results

We evaluate XNet across three domains—function approximation, PDE solving, and reinforcement learning—selected to assess different capabilities of neural architectures. These tasks respectively test expressivity under discontinuities, accuracy in modeling physical operators, and generalization in sequential decision-making.

This diversity highlights XNet's key strengths: high-order approximation, compact design, and efficient training. In all settings, we compare against MLP and KAN baselines under matched parameter budgets and consistent training protocols.

## 4.1 Function Approximation Experiments

We evaluate the networks on four categories of functions with increasing complexity: a low-dimensional discontinuous function, low-dimensional special functions, high-dimensional functions, and a function with noise. The experimental comparison between XNet, B-spline, and KAN indicates that XNet provides enhanced approximation capabilities. Except for the first function example and the final task, all other examples are from the referenced article [19], with KAN settings matching those from the original experiments. This ensures a fair comparison, demonstrating that XNet exhibits strong approximation capabilities across various benchmarks.

### 4.1.1 Heaviside step function approximation

**Dataset and Implementation Details.**   We evaluate discontinuous function approximation capabilities using the Heaviside step function:

$$f(x) = \begin{cases} 1, & x > 0, \\ 0, & x \leq 0. \end{cases} \tag{1}$$

An XNet architecture [1, 64, 1] and a KAN configuration [1, 1] with 200 grids were implemented. Both models were trained using the Adam optimizer [12] under identical conditions.

Table 3: Heaviside function approximation comparison

| Model | Architecture | MSE | RMSE | MAE |
|---|---|---|---|---|
| XNet | [1,64,1] | **8.99e-08** | **3.00e-04** | **1.91e-04** |
| ReLU (shallow) | [1,64,1] | 2.05e-03 | 4.53e-02 | 8.82e-03 |
| ReLU (deep) | [1,64,64,1] | 6.81e-05 | 8.25e-03 | 3.76e-04 |
| KAN | [1,1] (200 grids) | 5.98e-04 | 2.45e-02 | 3.03e-03 |

**Overall Performance.**   As shown in Figure 4, both B-Spline and KAN exhibit "overshoot," resulting in local oscillations at discontinuities. While adjusting the grid can mitigate this phenomenon, it introduces complexity in parameter tuning (see Table 9, Appendix B.2). In contrast, XNet demonstrates superior performance, providing smoother transitions at discontinuities. In terms of fitting accuracy in these regions, XNet outperforms KAN, with an MSE approximately 1000-fold smaller (8.99e-08 versus 5.98e-04) as shown in Table 3. The mathematical foundation for XNet's advantages in discontinuous functions is provided in Section 3.2.

### 4.1.2 Approximation with special functions

**Dataset and Implementation Details.**   We collect several special functions common in math and physics, summarized in Table 4. For neural network architectures, we implemented consistent parameter configurations following [19]. MLPs are configured with fixed widths of 5 or 100, with depths varying across $\{2, 3, 4, 5, 6\}$. KANs maintain a fixed width of 5 with depths similarly swept through $\{2, 3, 4, 5, 6\}$. XNet was constructed as a single-layer architecture with either 500 or 5000 basis functions.

Table 4: Special functions benchmark comparison with fixed XNet shape and extended test RMSEs.

| Name | scipy.special API | Minimal KAN shape | XNet shape | Best KAN test RMSE | MLP test RMSE | XNet test RMSE |
|---|---|---|---|---|---|---|
| Jacobian elliptic functions | ellipj(x, y) | [2,2,1] | [2,500,1] | $1.33e-04$ | $6.48e-04$ | $\mathbf{4.03e-05}$ |
| Incomplete elliptic integral of the first kind | ellipkinc(x, y) | [2,2,1,1] | [2,500,1] | $1.24e-04$ | $5.52e-04$ | $\mathbf{6.76e-05}$ |
| Incomplete elliptic integral of the second kind | ellipeinc(x, y) | [2,2,1,1] | [2,500,1] | $8.26e-05$ | $3.04e-04$ | $\mathbf{5.60e-05}$ |
| Bessel function of the first kind | jv(x, y) | [2,3,1,1,1] | [2,500,1] | $1.64e-03$ | $5.52e-03$ | $\mathbf{2.39e-04}$ |
| Bessel function of the second kind | yv(x, y) | [2,2,2,1] | [2,5000,1] | $1.49e-05$ | $3.45e-04$ | $\mathbf{1.12e-05}$ |
| Modified Bessel function of the first kind | iv(x, y) | [2,4,3,2,1,1] | [2,500,1] | $9.28e-03$ | $1.07e-02$ | $\mathbf{6.39e-05}$ |
| Associated Legendre function ($m = 0$) | lpmv(0, x, y) | [2,2,1] | [2,500,1] | $5.25e-05$ | $1.74e-02$ | $\mathbf{4.01e-05}$ |
| Associated Legendre function ($m = 1$) | lpmv(1, x, y) | [2,4,1] | [2,500,1] | $6.90e-04$ | $1.50e-03$ | $\mathbf{2.71e-04}$ |
| Associated Legendre function ($m = 2$) | lpmv(2, x, y) | [2,3,2,1] | [2,500,1] | $2.26e-04$ | $9.43e-04$ | $\mathbf{7.44e-05}$ |
| spherical harmonics ($m = 1, n = 1$) | sph_harm(1,1,x,y) | [2,3,2,1] | [2,500,1] | $1.22e-04$ | $6.70e-04$ | $\mathbf{2.28e-05}$ |
| spherical harmonics ($m = 1, n = 2$) | sph_harm(1,2,x,y) | [2,2,1,1] | [2,500,1] | $1.50e-05$ | $1.84e-03$ | $\mathbf{7.78e-06}$ |
| spherical harmonics ($m = 2, n = 2$) | sph_harm(2,2,x,y) | [2,2,3,2,1] | [2,500,1] | $9.45e-05$ | $6.21e-04$ | $\mathbf{2.43e-05}$ |

**Overall Performance.** As presented in Table 4, experiments demonstrate that XNet achieves superior approximation accuracy on the special functions. Particularly noteworthy are XNet's results on complex functions such as spherical harmonics, where it achieves an RMSE of $7.78 \times 10^{-6}$ for $(m = 1, n = 2)$ compared to KAN's $1.50 \times 10^{-5}$ and MLP's $1.84 \times 10^{-3}$. For Bessel functions, XNet similarly shows considerable improvement, with an RMSE of $2.39 \times 10^{-4}$ for the first kind versus $1.64 \times 10^{-3}$ and $5.52 \times 10^{-3}$ for KAN and MLP respectively. These findings suggest that XNet's single-layer architecture effectively approximates special functions with high accuracy.

### 4.1.3 Approximation with high-dimensional functions

**Dataset and Implementation Details.** Following the benchmarks established in [19], we evaluated two challenging functions: a 4-dimensional function $\exp\left(\frac{1}{2}\left(\sin\left(\pi(x_1^2 + x_2^2)\right) + x_3 x_4\right)\right)$ and a 100-dimensional function $\exp\left(\frac{1}{100}\sum_{i=1}^{100}\sin^2\left(\frac{\pi x_i}{2}\right)\right)$. Each experiment utilized 8000 training points and 1000 test points. For the 4D function, KAN was configured as $[4, 4, 2, 1]$ and XNet was implemented with 5000 basis functions. For the 100D function, KAN was structured as $[100, 1, 1]$ with XNet maintaining the same configuration. XNet was optimized using the Adam algorithm, while KAN is initialized with $G = 3$ grid points and trained with the LBFGS algorithm [18], progressively increasing grid density through $G = 3, 5, 10, 20, 50$ every 200 steps.

Table 5: Performance comparison of XNet and KAN on 4D and 100D test functions

| Function | Model | MSE | RMSE | MAE | Time (s) |
|---|---|---|---|---|---|
| $e^{\frac{1}{2}\left(\sin\left(\pi(x_1^2 + x_2^2)\right) + x_3 x_4\right)}$ | XNet [2, 5000, 1] | 2.31e-06 | 1.52e-03 | 8.39e-04 | 78.18 |
| | KAN [4,2,2,1] | 2.62e-03 | 5.11e-02 | 3.63e-02 | 143.10 |
| $e^{\frac{1}{100}\sum_{i=1}^{100}\sin^2\left(\frac{\pi x_i}{2}\right)}$ | XNet [2, 5000, 1] | 6.85e-04 | 2.62e-02 | 2.09e-02 | 158.69 |
| | KAN [100, 1, 1] | 6.59e-03 | 8.12e-02 | 6.46e-02 | 556.50 |

**Overall Performance.** The results presented in Table 5 indicate that XNet outperforms KAN across both dimensionalities. For the 4D function, XNet achieves an MSE three orders of magnitude lower than KAN while requiring approximately half the computation time. In the challenging 100D case, XNet maintains nearly an order of magnitude better MSE while executing 3.5× faster than KAN. Notably, XNet's computational efficiency scales significantly better as dimensionality increases, while achieving consistently higher accuracy across all evaluation metrics.

### 4.1.4 Functions with noise

**Dataset and Implementation Details.** To assess function approximation capabilities in noisy environments, we examined a dynamic system governed by:

$$x_5^i = 0.1 x_0^i x_1^i + 0.5 \sin(x_2^i x_3^i) + \sin(x_4^i) + \mu^i,$$

where $i = 1, 2, ..., n$, with state transitions:

$$x_0^i = x_1^{i-1}, x_1^i = x_2^{i-1}, x_2^i = x_3^{i-1}, x_4^i = x_5^{i-1}.$$

Initial conditions $x_0^0, x_1^0, x_2^0, x_3^0, x_4^0$ were sampled from $[0, 0.2]$, with Gaussian noise $\mu^i \sim N(0, \sigma^2)$ at three distinct levels: none ($\sigma = 0$), moderate ($\sigma = 0.05$), and high ($\sigma = 0.1$). We generated 300 sequential points, training models on the first $80\%$ (noise-corrupted data) and evaluating prediction performance on the remaining $20\%$ (noise-free data). This experimental design evaluates each model's capability to extract underlying patterns despite noise in the training data. For this experiment, we compared KAN [5, 64, 1] against XNet [5, 20, 1], both optimized using the Adam optimizer.

**Overall Performance.** Table 6 summarizes the comparative results. XNet outperforms KAN across all metrics, particularly in noisy environments. In the noise-free scenario, XNet achieves an MSE of $4.7099 \times 10^{-7}$, approximately 28× lower than that of KAN ($1.3209 \times 10^{-5}$). As noise increases, XNet maintains its advantage, with 3.7× better MSE at moderate noise and 14.5× better at high noise. Figures 8 and 9 provide visual evidence of XNet's enhanced robustness to noise. Furthermore, XNet achieves these performance improvements with fewer parameters and reduced training times. These results indicate that XNet may be more suitable for real-world noisy datasets, while the KAN model exhibits characteristics consistent with overfitting.

Table 6: Performance comparison of KAN [5, 64, 1] and XNet [5, 20, 1] under different noise levels.

| Noise Level | Model | MSE | RMSE | MAE | Time (s) |
|---|---|---|---|---|---|
| 0 | **KAN [5, 64, 1]** | 1.3209e-05 | 3.6344e-03 | 4.8311e-03 | 17.8863 |
| | **XNet [5, 20, 1]** | 4.7099e-07 | 6.8629e-04 | 5.2943e-04 | 9.6502 |
| 0.05 | **KAN [5, 64, 1]** | 1.6784e-03 | 4.0968e-02 | 3.1549e-02 | 17.9535 |
| | **XNet [5, 20, 1]** | 4.5109e-04 | 2.1239e-02 | 1.5797e-02 | 9.4391 |
| 0.1 | **KAN [5, 64, 1]** | 9.4653e-03 | 9.7290e-02 | 7.8063e-02 | 17.8863 |
| | **XNet [5, 20, 1]** | 6.5517e-04 | 2.5596e-02 | 2.0233e-02 | 9.0296 |

Given its performance in function approximation tasks, both in terms of computational efficiency and accuracy, experimental results suggest that XNet provides an efficient neural network architecture with effective approximation capabilities. Building on this, in the following subsection, we apply PINN, KAN, and XNet to approximate the value function of the Heat and Poisson equations.

## 4.2 PDE Solving Capability

**Experimental Setup and Implementation Details.** We selected the Heat equation as a benchmark PDE problem:

$$\frac{\partial u}{\partial t} = \nu \frac{\partial^2 u}{\partial x^2}, \quad (x, t) \in [0, 1] \times [0, 1], u(x, 0) = \sin(\pi x), \quad x \in [0, 1], u(0, t) = u(1, t) = 0, \quad (2)$$

with known analytical solution $u(x, t) = e^{-\nu \pi^2 t} \sin(\pi x)$. Following the physics-informed neural networks (PINNs) framework, we defined the loss function as

$$\text{loss} = \alpha \, \text{loss}_i + \text{loss}_o = \frac{1}{n_o} \sum_{i=1}^{n_o} \left| u^\theta(z_i) - u(z_i) \right|^2 + \alpha \frac{1}{n_i} \sum_{i=1}^{n_i} \left| u_t^\theta(z_i) - \nu u_{xx}^\theta(z_i) \right|^2, \quad (3)$$

where $n_i = 2500$ interior points and $n_o = 150$ boundary points were sampled uniformly, with $\alpha = 0.1$ serving as a weighting coefficient to balance the loss components.

We evaluate four network architectures within the PINN framework: a two-layer MLP $[2, 20, 20, 1]$, a single-layer KAN $[2, 10, 1]$, and two XNet configurations with varying widths $[2, 20, 1]$ and $[2, 200, 1]$.

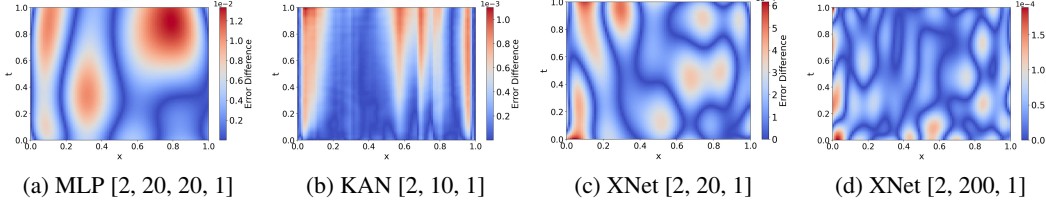

(a) MLP [2, 20, 20, 1]    (b) KAN [2, 10, 1]    (c) XNet [2, 20, 1]    (d) XNet [2, 200, 1]

Figure 2: Performance comparison of different neural network architectures

Table 7: Comparison of XNet and KAN on the Heat equation.

| Metric | MSE | RMSE | MAE | Time (s) |
|---|---|---|---|---|
| **MLP [2,20,20,1]** | 2.4536e-05 | 4.9534e-03 | 3.8323e-03 | 43.8 |
| **XNet [2,20,1]** | 3.8936e-08 | 1.9732e-04 | 1.5602e-04 | 43.5 |
| **KAN [2,10,1]** | 1.5106e-07 | 3.8866e-04 | 2.9661e-04 | 254.6 |
| **XNet [2,200,1]** | 3.6867e-09 | 6.0718e-05 | 5.0027e-05 | 108.3 |

**Overall Performance.** Table 7 quantitatively compares the performance of MLP, XNet, and KAN architectures on the heat equation task. Across all evaluated metrics, XNet consistently achieves superior accuracy and computational efficiency. The [2,200,1] XNet configuration achieves the lowest error, with an MSE of $3.69 \times 10^{-9}$—approximately two orders of magnitude lower than

KAN ($1.51 \times 10^{-7}$)—while also requiring less than half the training time (108.3s vs. 254.6s). Even with a more compact architecture ([2,20,1]), XNet outperforms KAN, showing a threefold lower MSE and completing training more than five times faster. In comparison, the standard MLP ([2,20,20,1]) produces higher errors across all metrics, highlighting the limitations of conventional architectures for PDE approximation. These results demonstrate XNet's effectiveness in capturing the underlying structure of PDE solutions, suggesting its promise as a compact and efficient model reduction framework.

Additionally, we evaluated physics-informed machine learning models based on MLP, KAN, and XNet architectures for solving the 2D Poisson equation (detailed results in Appendix B.6). The XNet [2,200,1] configuration outperforms KAN [2,10,1] in both accuracy and computational efficiency, while achieving approximately $20,000\times$ better precision compared to the standard PINN [2,20,20,1] architecture.

## 4.3 Reinforcement Learning Applications

**Experimental Setup and Implementation Details.** We evaluated XNet, KAN, and MLP architectures as function approximators within the Proximal Policy Optimization (PPO) framework on two continuous control tasks from the DeepMind Control Suite: *HalfCheetah-v4* and *Swimmer-v4*.

Proximal Policy Optimization (PPO) optimizes the stochastic policy $\pi_\theta$ by employing a clipped surrogate objective function to ensure stable policy updates:

$$L_{\text{PPO}}(\theta) = \mathbb{E}_t \left[ \min \left( r_t(\theta)\hat{A}_t, \text{clip}(r_t(\theta), 1 - \epsilon, 1 + \epsilon)\hat{A}_t \right) \right], \tag{4}$$

where $r_t(\theta)$ represents the policy ratio and $\epsilon$ is a clipping parameter (configured at 0.2). This objective function is combined with a value function loss and entropy regularization to balance exploration and exploitation.

Table 8 summarizes the network configurations used in our experiments. The MLP baseline uses two hidden layers with 64 units each for both actor and critic networks. The KAN implementation employs spline functions with order $k = 2$ and grid size $g = 3$, chosen based on preliminary work [11]. The XNet configuration utilizes 64 basis functions to maintain parameter count parity with the MLP architecture for fair comparison. All models were trained using the Adam optimizer, implementing identical hyperparameters across architectures to isolate the effects of the function approximator choice.

Table 8: Network configurations for reinforcement learning tasks.

| Model | Configuration |
|---|---|
| **MLP** | Two hidden layers with 64 units each |
| **KAN** | Order $k = 2$, grid size $g = 3$ |
| **XNet** | 64 basis functions |

**Overall Performance.** Figure 3 summarize the results of the three models on the two tasks. XNet consistently outperforms MLP and KAN in both environments, achieving the highest rewards and faster convergence. In *HalfCheetah-v4* environment, XNet improves the reward by 64% compared to KAN and 142% compared to MLP. In *Swimmer-v4* environment, XNet achieves reward improvements of 47% over MLP and 28% over KAN.

**Theoretical Analysis of XNet's Advantage.** XNet's performance in reinforcement learning stems from its Cauchy kernel activation, which provides three key benefits:

1. **Localized response with fast decay:** Cauchy activations enable sharper adaptation to localized patterns in policy and value functions—particularly effective in continuous control tasks with abrupt transitions.

2. **High-order approximation capability:** As shown in Theorem, XNet supports arbitrarily high-order convergence for analytic targets, yielding more precise representations than MLPs or KANs.

3. **Reduced parameter complexity:** XNet achieves strong performance with fewer parameters due to its expressive basis functions, leading to faster convergence in reinforcement learning.

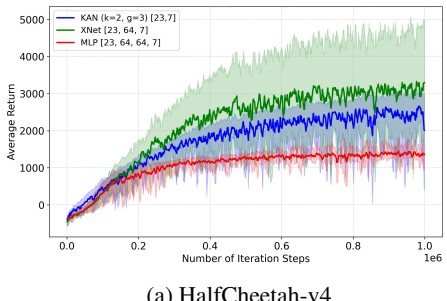

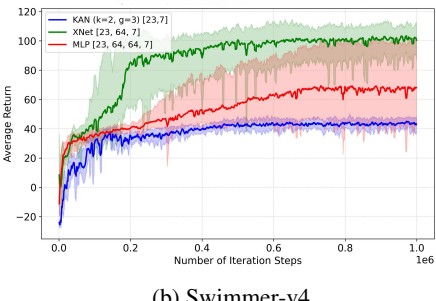

|(a) HalfCheetah-v4|(b) Swimmer-v4|

Figure 3: Reward comparison for PPO training across environments. XNet significantly outperforms both KAN and MLP in final performance after 1M training steps: In HalfCheetah-v4, XNet achieves 3298.52, compared to KAN (2010.52) and MLP (1358.49). In Swimmer-v4, XNet reaches 100.38, while KAN and MLP attain 43.25 and 68.08 respectively.

These properties align well with the demands of policy gradient methods like PPO, where accurate approximation of both policy $\pi_\theta(a|s)$ and value function $V(s)$ is crucial. Figure 3 empirically confirms these advantages.

## 5 Conclusion and Outlook

Our results suggest that neural expressivity need not come from depth alone. XNet, built on principled analytic activation functions, achieves high-order approximation with only a single layer—offering both theoretical rigor and empirical power.

This design unlocks a new perspective for compact and interpretable architectures, particularly in scientific and symbolic domains. Its ability to approximate discontinuous functions, solve PDEs efficiently, and improve sample efficiency in RL agents makes it a promising candidate for hybrid symbolic-numeric solvers and real-time control systems.

In future work, we plan to integrate XNet into multi-scale or Transformer-style models, extend it to generative and multi-modal learning, and explore its role in operator learning and symbolic program induction.

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

# A Theoretical Foundations

## A.1 Computational Efficiency Analysis

**FLOP Comparison:** The Cauchy activation $\phi(x) = \frac{\lambda_1 x + \lambda_2}{x^2 + d^2}$ introduces 2-3× more FLOPs per activation compared to ReLU, but remains significantly lighter than spline-based KANs.

**Time-to-Solution:** Despite higher per-activation cost, XNet achieves faster convergence. For example, on the Heat equation (Table 7), XNet [2,200,1] reaches target accuracy in 108s vs KAN's 254s, demonstrating that expressivity gains outweigh computational overhead.

**Scalability:** For Transformer architectures, Cauchy maintains $O(n)$ complexity where $n$ is sequence length, adding only a constant factor increase in FLOPs.

## A.2 Approximation Efficiency

We recall the classical approximation rate result from Cauchy kernel expansions. If $f \in C^p(K)$, then for any $p > 0$, there exists a rational function $f_N$ composed of $N$ Cauchy basis functions such that:

$$\|f - f_N\|_\infty \le C(p)N^{-p},$$

where $C(p)$ depends on the smoothness of $f$ and the domain geometry.

To achieve error at most $\epsilon$, we require:

$$N = O\left(\epsilon^{-1/p}\right).$$

In contrast, ReLU networks yield $O(N^{-1/d})$ for $d$-dimensional $f$, and KANs offer $O(N^{-k})$ bounded by spline degree $k$. Thus, XNet provides tunable convergence rates with fewer neurons for smooth targets.

## A.3 Localization and Adaptivity

The activation function:

$$\phi(x) = \frac{\lambda_1 x + \lambda_2}{x^2 + d^2}$$

satisfies:

$$\lim_{|x| \to \infty} \phi(x) = 0, \quad \phi'(x) = \frac{\lambda_1(x^2 + d^2) - 2x(\lambda_1 x + \lambda_2)}{(x^2 + d^2)^2}.$$

For localized representation centered at $\mu_i$, we define:

$$\phi_i(x) = \frac{\lambda_{1,i}(x - \mu_i) + \lambda_{2,i}}{(x - \mu_i)^2 + d_i^2}.$$

This enables each neuron to adapt to regions of discontinuity or local variation. Unlike B-splines, XNet does not require explicit partitioning or fixed grid support.

## A.4 Matrix Conditioning and Derivative Properties

**Matrix Conditioning:** The kernel matrix $A$ with entries $A_{ij} = \phi_i(x_j)$ is dense. Using Gershgorin's theorem:

$$|\lambda - A_{ii}| \le \sum_{j \ne i} |A_{ij}|,$$

the condition number scales as $\kappa(A) = O(N)$ in general, potentially requiring preconditioning for large $N$. B-spline matrices are sparse and banded with $\kappa(A) = O(1)$.

**Derivative Computation:** For efficient optimization, we compute:

$$\frac{d^n}{d(d^2)^n}\left(\frac{1}{x^2+d^2}\right) = (-1)^n \frac{(n-1)!}{(x^2+d^2)^{n+1}}.$$

Cauchy activations are fully differentiable with respect to $d^2$, enabling second-order or adaptive methods.

## A.5  Proof of High-Order Approximation

**Theorem 3.2** (High-Order Approximation of XNet). *Let $K \subset \mathbb{R}^d$ be compact, and let $f \in C^\omega(K)$ be real analytic. Let $\Phi = \left\{\frac{\lambda_1 x + \lambda_2}{x^2 + d^2}\right\}$ be the Cauchy activation family. Then for any $r > 0$, there exists a one-hidden-layer XNet function $f_N \in span(\Phi)$ such that*

$$\|f - f_N\|_{C^0(K)} = O(N^{-r}).$$

*Proof.* We now establish the high-order approximation capability of XNet in the uniform norm $C^0(K)$ for any compact set $K \subset \mathbb{R}^d$. The proof is divided into three steps: we begin in the one-dimensional real setting, then extend to high-dimensional polynomials using a classical lifting theorem, and finally conclude by composing the approximations.

Step 1: One-dimensional high-order approximation via the Cauchy approximation theorem.

Let $I = [-1, 1] \subset \mathbb{R}$, and let $f \in C^\omega(I)$ be any real-analytic function on the interval. By the one-dimensional Cauchy approximation theorem (a special case of Theorem 1 in [16]), for any $\varepsilon > 0$, $\left\|f(x) - \sum_{k=1}^{M}\frac{\lambda_k}{\xi_k - x}\right\|_{C^0(I)} < \varepsilon$, i.e., we have $g_M(x) = \sum_{k=1}^{M}\phi_k(x)$:

$$\|f - g_M\|_{C^0(I)} = O(M^{-r}) \quad \text{for any } r > 0.$$

Step 2: Multivariate polynomial approximation by Cauchy activations (XNet)

Let $K \subset \mathbb{R}^d$ be a compact set, and let $\mathbb{P}_n(\mathbb{R}^d)$ denote the set of all real multivariate polynomials of total degree at most $n$.

We refer to the General Approximation Theorem in [16]. If the one-dimensional function $\phi(x)$ can approximate all monomials $x^k$ with arbitrarily high-order accuracy in the $C^0$ norm, then for any fixed $w \in \mathbb{R}^d$, the functions $\phi(w^\top x + b)$ can approximate $(w^\top x)^k$ with the same rate. Since any multivariate monomial $x^\alpha = \prod_{j=1}^{d}(x^j)^{\alpha_j}$ can be written as a linear combination of such functions (as shown in [16]), it follows that Cauchy activations of the form $\phi(w^\top x + b)$ can approximate $x^\alpha$ with arbitrarily high-order convergence in $C^0(K)$.

Step 3: High-order approximation of analytic functions via XNet

Let $f \in C^\omega(K)$ be a real analytic function on a compact domain $K \subset \mathbb{R}^d$. By classical approximation theory (e.g., Bernstein, Mhaskar), there exists a sequence of multivariate polynomials $\{p_n(x)\} \subset \mathbb{P}_n(\mathbb{R}^d)$ such that

$$\|f - p_n\|_{C^0(K)} = O(n^{-r}) \quad \text{for any } r > 0.$$

From Step 2, each polynomial $p_n$ can be approximated by a function $g_M$ in the XNet class:

$$g_M(x) = \sum_{i=1}^{M} a_i \cdot \phi(w_i^\top x + b_i),$$

with

$$\|p_n - g_M\|_{C^0(K)} = O(M^{-s}) \quad \text{for any } s > 0.$$

Applying the triangle inequality, we obtain:

$$\|f - g_M\|_{C^0(K)} \leq \|f - p_n\|_{C^0(K)} + \|p_n - g_M\|_{C^0(K)} = O(n^{-r}) + O(M^{-s}).$$

By choosing $M = n$, we deduce:

$$\|f - g_M\|_{C^0(K)} = O(M^{-k}) \quad \text{for any } k > 0.$$

Conclusion:

Let $\phi(x) = \frac{\lambda_1 x + \lambda_2}{x^2 + d^2}$, and define the single-layer neural network class (XNet) as

$$\mathcal{X}_M := \left\{ x \mapsto \sum_{i=1}^{M} a_i \cdot \phi(w_i^\top x + b_i) \right\}.$$

Then for any compact set $K \subset \mathbb{R}^d$, and any function $f \in C^\omega(K)$, there exists $g_M \in \mathcal{X}_M$ such that

$$\|f - g_M\|_{C^0(K)} = O(M^{-k}) \quad \text{for any } k > 0.$$

That is, XNet achieves arbitrarily high-order approximation accuracy for real analytic functions in the uniform norm.

$\square$

## A.6  Neuron Efficiency under Arbitrary Approximation Orders

The table in Section 3.2 summarizes how different architectures compare in terms of neurons required to reach a desired approximation error $\epsilon$. Here, we provide the derivation behind the scaling laws.

**XNet.**  From Theorem 3.2, we have

$$\|f - f_N\| \leq C N^{-r} \Rightarrow N = O(\epsilon^{-1/r}).$$

This result holds for arbitrary $r > 0$, assuming $f \in C^k(K)$ and sufficient smoothness.

**ReLU MLP.**  For classical ReLU networks [1], the best known approximation rate is:

$$\|f - f_N\| \leq C N^{-1/d} \Rightarrow N = O(\epsilon^{-d}),$$

which reflects the curse of dimensionality.

**KAN.**  For B-spline-based KANs [19], convergence is limited by the spline degree $k$:

$$\|f - f_N\| \leq C N^{-k} \Rightarrow N = O(\epsilon^{-1/k}),$$

and the degree $k$ is typically small (e.g., $k = 3$), fixed by construction.

**Summary.**  These derivations support the theoretical table in Section 3.2, confirming that XNet allows flexible approximation with significantly fewer neurons under high-order settings.

# B  Experimental Details

## B.1  Implementation Environment

The numerical experiments presented below were performed in Python using the Tensorflow-CPU processor on a Dell computer equipped with a 3.00 Gigahertz (GHz) Intel Core i9-13900KF. When detailing grids ans k for KAN models, we always use values provided by respective authors (Kan).

## B.2  One-Dimensional Discontinuous FUNCTION APPROXIMATION

Figure 4 provides visualizations of approximation results for the Heaviside function, highlighting the characteristic "overshoot" phenomenon in B-spline and KAN methods near the discontinuity, which XNet successfully avoids.

Table 9 presents an extensive comparison of various KAN configurations, showing that even with deeper architectures ([1,3,1]), KAN cannot match XNet's approximation accuracy for discontinuous functions. The B-spline results further demonstrate that increasing the spline degree $k$ from 3 to 10 yields only marginal improvements, while XNet provides orders of magnitude better approximation with its single-layer architecture.

Table 9: KAN reference

| k,G | [1,1]KAN | | | [1,3,1]KAN | | |
|---|---|---|---|---|---|---|
| | MSE | RMSE | MAE | MSE | RMSE | MAE |
| k=3, G=3 | 2.20E-02 | 1.48E-01 | 9.89E-02 | 3.50E-04 | 1.87E-02 | 5.56E-03 |
| k=3, G=10 | 1.22E-02 | 1.10E-01 | 5.91E-02 | 1.84E-04 | 1.36E-02 | 2.54E-03 |
| k=3, G=50 | 2.44E-03 | 4.94E-02 | 1.22E-02 | 4.28E-05 | 6.55E-03 | 2.71E-03 |
| k=3, G=200 | 5.98E-04 | 2.45E-02 | 3.03E-03 | 3.79E-04 | 1.95E-02 | 1.24E-02 |

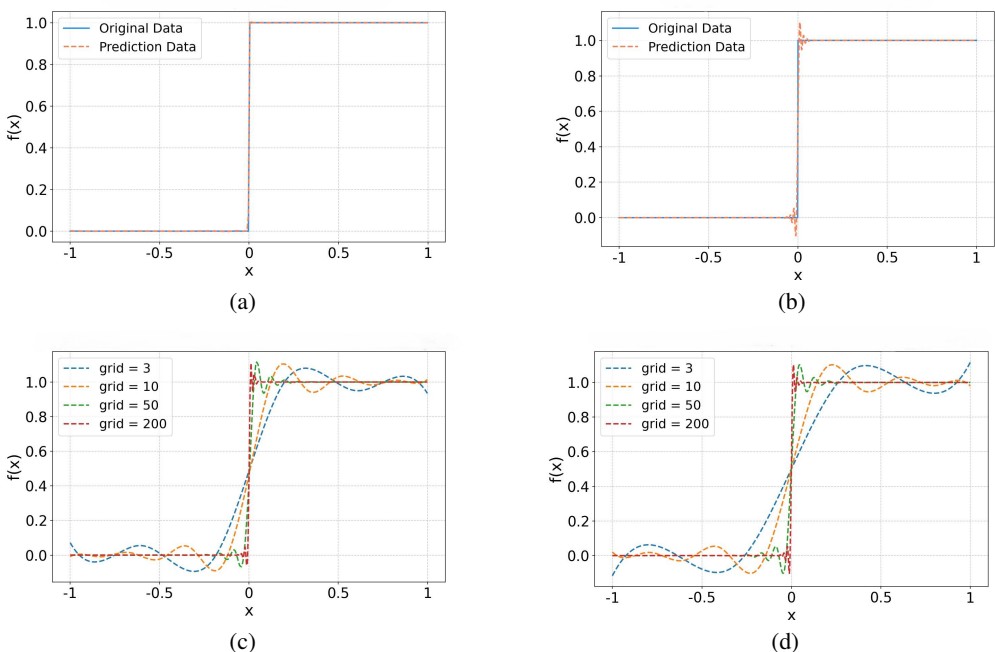

Figure 4: Heaviside step function approximation comparison: (a) XNet, with 64 basis functions. (b) KAN [1,1], with $k = 3$, grid = 200. (c) B-Spline, with $k = 3$. (d) KAN [1,1], with $k = 3$.

Table 10: B-Spline Performance metrics comparison for different G and K values. reference

| B-Spline | | | |
|---|---|---|---|
| k, G | MSE | RMSE | MAE |
| k=50, G=200 | 5.8477e-01 | 7.6470e-01 | 6.1076e-01 |
| k=3, G=10 | 9.2871e-03 | 9.6369e-02 | 4.7923e-02 |
| k=3, G=50 | 2.3252e-03 | 4.8221e-02 | 1.2255e-02 |
| k=10, G=50 | 1.9881e-03 | 4.4588e-02 | 1.0879e-02 |
| k=3, G=200 | 1.1252e-03 | 3.3544e-02 | 4.4737e-03 |
| k=10, G=200 | 1.1029e-03 | 3.3210e-02 | 5.1904e-03 |

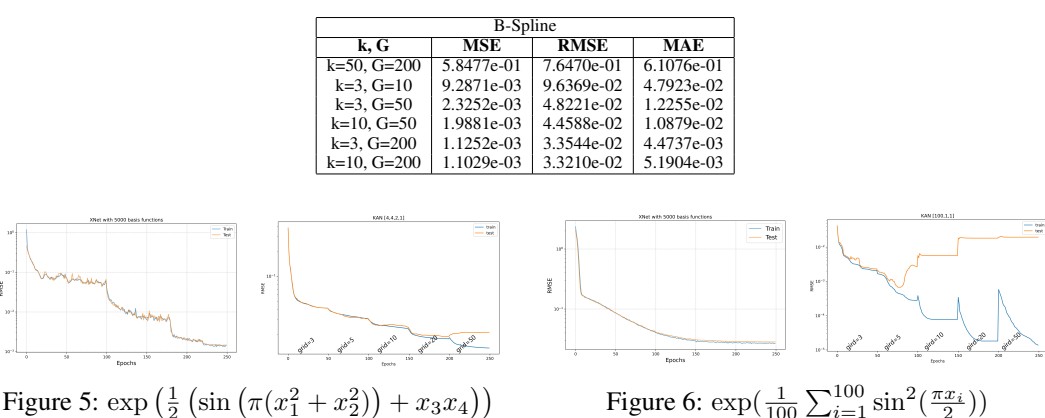

Figure 5: $\exp\left(\frac{1}{2}\left(\sin\left(\pi(x_1^2 + x_2^2)\right) + x_3 x_4\right)\right)$

Figure 6: $\exp\left(\frac{1}{100}\sum_{i=1}^{100}\sin^2\left(\frac{\pi x_i}{2}\right)\right)$

## B.3 High-Dimensional Continuous FUNCTION APPROXIMATION

Figure 5 and 6 show the training loss curves for both XNet and KAN on the 4D and 100D test functions.

Figure 7 shows XNet's performance with varying numbers of Cauchy basis functions. Performance improves consistently with increasing width, confirming the theoretical scaling laws.

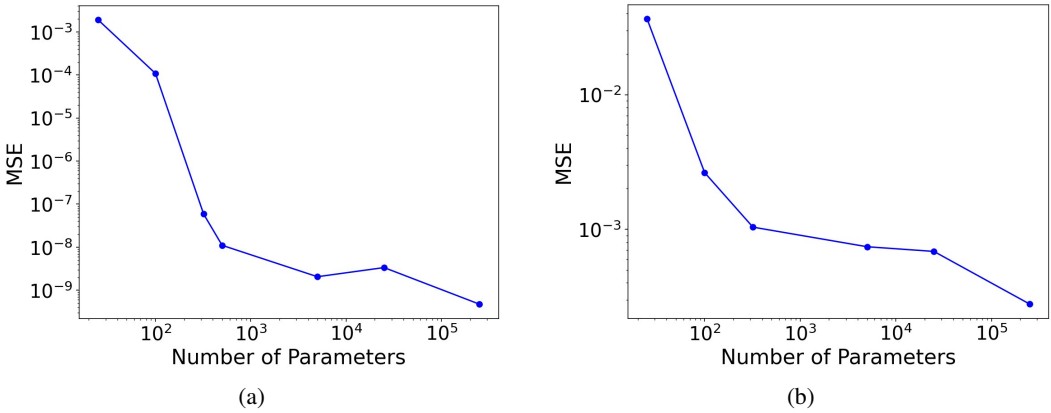

(a)

(b)

Figure 7: Performance of XNet on approximating different functions with varying numbers of parameters: (a) $\exp\left(\frac{1}{2}\left(\sin\left(\pi(x_1^2 + x_2^2)\right) + x_3 x_4\right)\right)$; and (b) $\exp\left(\frac{1}{100}\sum_{i=1}^{100}\sin^2\left(\frac{\pi x_i}{2}\right)\right)$.

## B.4 Function Approximation with noise

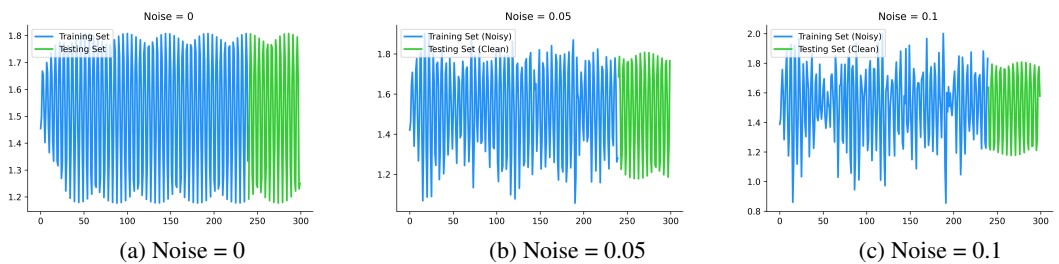

(a) Noise = 0      (b) Noise = 0.05      (c) Noise = 0.1

Figure 8: Function fitting tested on datasets with different noise levels: (a) Noise = 0, (b) Noise = 0.05, and (c) Noise = 0.1. Blue lines represent the noise-free ground truth function, while red dots represent the noisy training samples.

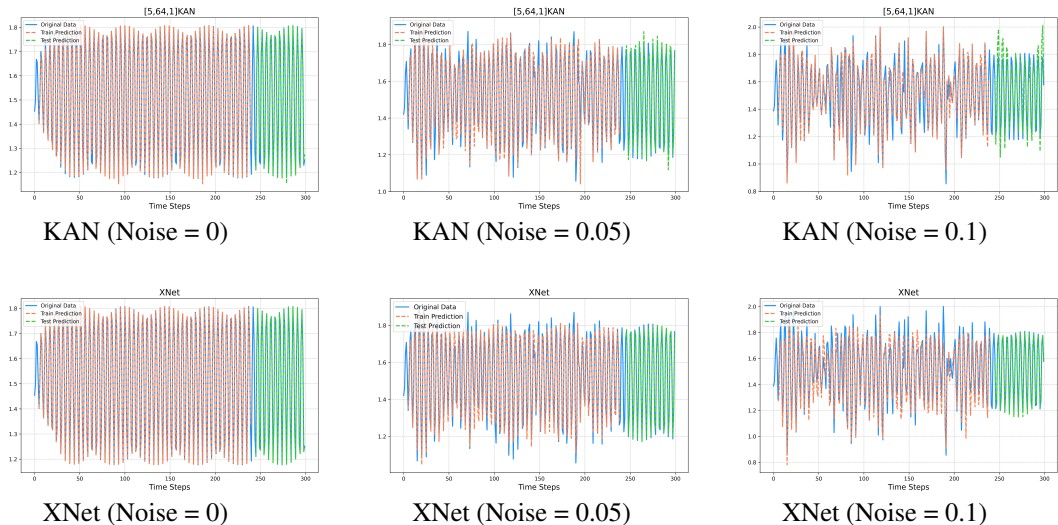

KAN (Noise = 0)      KAN (Noise = 0.05)      KAN (Noise = 0.1)

XNet (Noise = 0)      XNet (Noise = 0.05)      XNet (Noise = 0.1)

Figure 9: Comparison of the performance of KAN and XNet under different noise levels. Blue lines represent ground truth, while red lines show predictions. Note that XNet better captures the underlying pattern even at high noise levels, while KAN shows signs of overfitting.

Figures 8 and 9 provide a detailed comparison of how XNet and KAN handle increasing noise levels. XNet significantly outperforms KAN across all noise levels, with the advantage becoming more pronounced as noise increases. This confirms XNet's theoretical noise robustness through adaptive localization. At the highest noise level, XNet achieves 14.5× lower MSE than KAN while requiring only half the training time.

## B.5 Heat equation

For solving the Heat equation under the framework of PINN, the loss function plots of various models (see Figure 10).

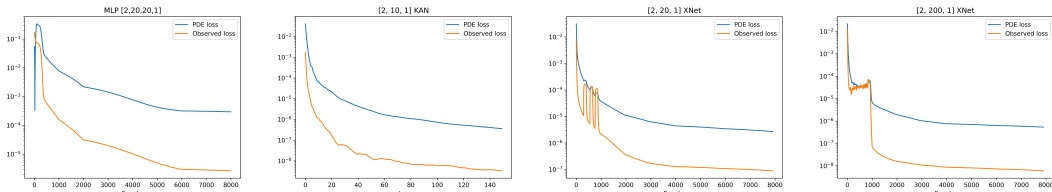

Figure 10: Comparison of KAN, PINN and XNet approximations on PDE loss.

## B.6 Poisson equation

**Task Description.** We aim to solve a 2D Poisson equation $\nabla^2 v(x, y) = f(x, y)$, $f(x, y) = -2\pi^2 \sin(\pi x)\sin(\pi y)$, with boundary condition $v(-1, y) = v(1, y) = v(x, -1) = v(x, 1) = 0$. The ground truth solution is $v(x, y) = \sin(\pi x)\sin(\pi y)$. We use the framework of physics-informed neural networks (PINNs) to solve this PDE, with the loss function given by

$$
\begin{aligned}
\text{loss}_{\text{pde}} &= \alpha \text{loss}_i + \text{loss}_b \\
&:= \alpha \frac{1}{n_i} \sum_{i=1}^{n_i} |v_{xx}(z_i) + v_{yy}(z_i) - f(z_i)|^2 + \frac{1}{n_b} \sum_{i=1}^{n_b} v^2 ,
\end{aligned}
\tag{5}
$$

where we use $\text{loss}_i$ to denote the interior loss, discretized and evaluated by a uniform sampling of $n_i$ points $z_i = (x_i, y_i)$ inside the domain, and similarly we use $\text{loss}_b$ to denote the boundary loss, discretized and evaluated by a uniform sampling of $n_b$ points on the boundary. $\alpha = 0.1$ is the hyperparameter balancing the effect of the two terms.

**Model Configurations.** We evaluate four network architectures within the PINN framework: a two-layer MLP $[2, 20, 20, 1]$, a single-layer KAN $[2, 10, 1]$, and two XNet configurations with different widths $[2, 20, 1]$ and $[2, 200, 1]$.

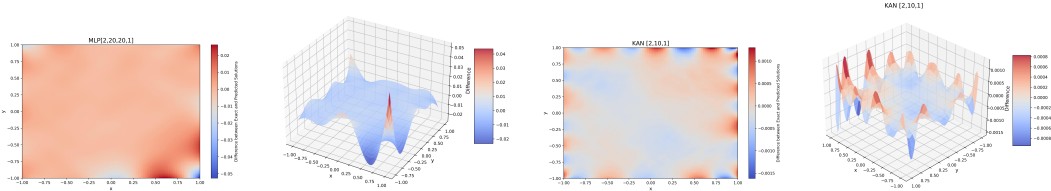

Figure 11: PINN and KAN Performance

Table 11: Comparison of XNet and KAN on the Poisson equation.

| Metric | MSE | RMSE | MAE | Time (s) |
|---|---|---|---|---|
| **PINN [2,20,20,1]** | 1.7998e-05 | 4.2424e-03 | 2.3300e-03 | 48.9 |
| **XNet (20)** | 1.8651e-08 | 1.3657e-04 | 1.0511e-04 | 57.2 |
| **KAN [2,10,1]** | 5.7430e-08 | 2.3965e-04 | 1.8450e-04 | 286.3 |
| **XNet (200)** | 1.0937e-09 | 3.3071e-05 | 2.1711e-05 | 154.8 |

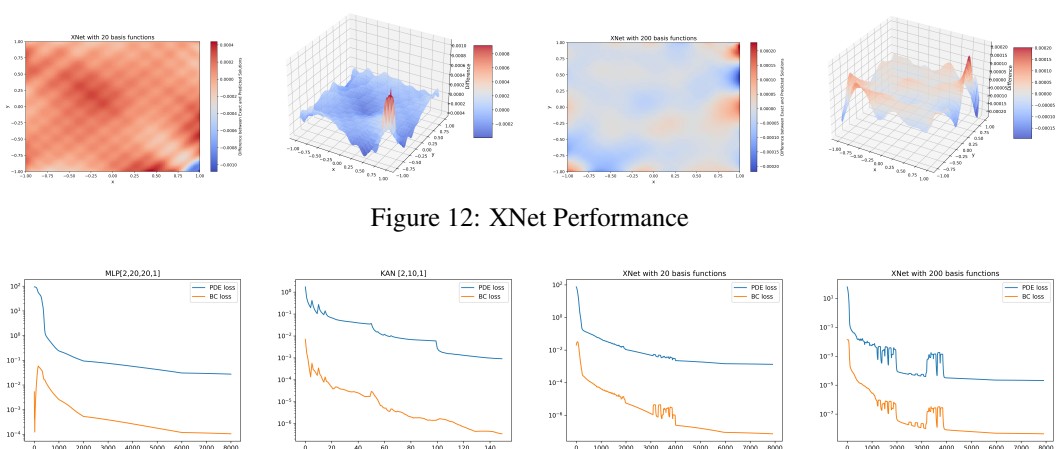

Figure 12: XNet Performance

Figure 13: Comparison of KAN, PINN and XNet on PDE loss. XNet shows smoother and faster convergence, with significantly lower final loss values than other methods.

**Comparative Results.** XNet-200 achieves approximately 53× lower MSE than KAN while requiring only 54% of the computational time. The more compact XNet-20 still outperforms KAN with 3× better accuracy and 5× faster training. Therefore we speculate that the XNet might have the potential of serving as a good neural network representation for model reduction of PDEs. In general, KANs and PINNs are good at representing different function classes of PDE solutions, which needs detailed future study to understand their respective boundaries.

## B.7 Benchmarking Study of different activation functions

This survey is compared with the existing survey/performance analysis and the experimental performance analysis of selected activation functions is performed over text data. The German to English translation is used to test the performance of the activation functions over text data. Benchmark Seq2Seq model consisting of a Long Short Term Memory (LSTM) based autoencoder network is used for the experiment. The model and dataset are downloaded from Kaggle2 [3].

Table 12: Experimental results for German to English language translation tasks.

| Activations | Bleu Score | Activations | Bleu Score |
|---|---|---|---|
| Cauchy | $21.47 \pm 0.67$ | Swish | $19.51 \pm 0.97$ |
| Sigmoid | $14.59 \pm 0.47$ | ABReLU | $17.55 \pm 0.63$ |
| Tanh | $20.93 \pm 0.91$ | LiSHT | $20.39 \pm 0.93$ |
| Elliott | $14.49 \pm 0.96$ | SRS | $20.66 \pm 0.78$ |
| ReLU | $18.88 \pm 0.86$ | Mish | $19.56 \pm 1.15$ |
| LReLU | $18.89 \pm 0.82$ | PAU | $20.11 \pm 1.24$ |
| PReLU | $20.04 \pm 0.98$ | Softplus | $16.78 \pm 0.84$ |
| ELU | $19.40 \pm 1.33$ | CELU | $18.71 \pm 0.55$ |
| SELU | $20.85 \pm 0.64$ | GELU | $18.75 \pm 1.83$ |
| SIRENs | $20.27 \pm 0.85$ | SiLU | $18.61 \pm 0.72$ |

Building upon the experiments conducted by Dubey et al. [6], we performed further updates and evaluations to assess the performance of different activation functions in language translation tasks. In these experiments, activation functions were applied to the feature embeddings immediately before the dropout layer. The training configuration included 50 epochs, a learning rate of 0.001, and a batch size of 256. Both the encoder and decoder had an embedding size of 300, with a dropout factor of 0.5. The Adam optimizer was employed, and training was conducted using cross-entropy loss. The Bleu score [21] with 4-gram evaluation was used as the metric for translation quality, with the mean and standard deviation of Bleu scores over five trials reported for each activation function. The results,

---

[3]https://www.kaggle.com/datasets/parthplc/translatedtext

summarized in Table 12, indicate that the Cauchy activation function consistently outperformed traditional activation functions such as Sigmoid, Tanh, and ReLU, achieving higher Bleu scores. This suggests that the Cauchy activation function is particularly well-suited for language translation tasks, as it enhances feature representation quality and demonstrates greater stability across training trials.

