# OpenReview forum: "From Kolmogorov to Cauchy: Shallow XNet Surpasses KANs"
_NeurIPS.cc/2025/Conference — NeurIPS 2025 poster_

### Official Review · Reviewer_wzmP · 2025-06-27

**Clarity:** 2
**Significance:** 4
**Originality:** 3
**Rating:** 5
**Confidence:** 3

**Summary:**

Finding a replacement for MLPs at the base of neural networks is an interesting problem to either increase computation efficiency or interpretability for simple networks, which was recently spearheaded by KANs. The authors introduce XNet, a neural network type based on the Cauchy basis functions, which has a theoretically superior approximation result for smooth functions. Experiments are provided to support this claim.

**Questions:**

- In XNet, do you need to tune the $p$ constant or is that automatically adapted to the highest smoothness order "detectable" from the target function? Is there any evidence if you claim the latter?
- In Table 5, is Time here training/inference time?

**Ethical Concerns:**

["NO or VERY MINOR ethics concerns only"]

**Final Justification:**

The authors have addressed all of my concerns. I have raised my score.

**Limitations:**

- The limitation about the assumption that the target function is infinitely differentiable (as mentioned in my review) is not discussed at all.
- Limitations about scaling XNet to larger networks should also be mentioned (is the memory usage or computation time a bottleneck).

**Quality:**

2

**Strengths And Weaknesses:**

## Strengths
- The problem of finding a provably better fundamental base for neural networks is an interesting and important question.
- The construction of XNet is novel given the previous work mentioned.
- The experimental results are good, although there are some concerns about fair comparison (see Weaknesses).
## Weaknesses
- It would be helpful to have a figure showing what Cauchy basis functions are. I believe this would make understanding the idea behind the paper more easily. As an example, the KAN paper has figures showing what the spline functions are and how the elementary operations over the spline functions operate in a layer.
- Misleading claim: The $p$ constant in the convergence rate for the Cauchy approximation result is the smoothness of the target function class. For most of the main text up to Section 3, this is never explicitly mentioned and it is only mentioned that the rate of $O(N^{-p})$ holds for arbitrary $p$. This is only true for infinitely differentiable target functions. I don't believe this is true for most applications we care about. This in turn makes it confusing why XNet is performing well in Section 4.1.1 because the target function is non-differentiable. Can the authors explain why XNet is performing well despite no theoretical guarantee for this particular case?
- Lack of clarity in writing: When introducing a new function like the Cauchy basis function, it is important to formally define what mathematical object it is. For example in Sec 3.2, it is not mentioned what the domain and co-domain of $\phi$ is.
- Missing baselines:
	- In Section 4.1.1, another baseline should be a comparable ReLU network. For non-differentiable targets, ReLU should be a natural fit and a more appropriate baseline.
- In Sec 4.1.2, the fairness in comparison between different configurations of the network width across different models is questionable. I am not familiar with the computation complexity of KAN and XNet, so I don't know if the much larger width in XNet (500 vs 100 in MLP and 6 in KAN) is fair. This is also true for the configuration in Table 8. Can you give an explanation for why you chose these hyperparams?

---

> ### Author Rebuttal · Authors · 2025-07-26
>
> We sincerely thank the reviewer for the thoughtful feedback and constructive suggestions. We will revise the text to more clearly clarify the scope of our contributions in relation to existing work, particularly [4].
>
> **Clarification of Theoretical Contribution:**
> [4] introduced the Cauchy function form and established general approximation theorems based on partial fraction decomposition, providing a theoretical foundation for Cauchy activation functions. However, their results are not specific to neural networks. In contrast, our **Theorem 3.2** provides the first **network-specific convergence guarantee**: we show that **XNet**, equipped with Cauchy activations, achieves **super-polynomial approximation rates** $O(N^{-r})$ for any $r$, assuming the target is real-analytic. This is a network-level result that requires fundamentally different analysis.
>
> This result goes beyond classical algebraic rates $O(N^{-r})$ and directly explains XNet’s empirical parameter efficiency, especially in Section 4.1.1. We will revise the main text to more clearly state the smoothness assumption (real-analyticity) and to clarify the distinction between our contribution and that of related work, including [4], to avoid further confusion.
> ## Response to Weaknesses
>
> ### W1: Visualization of Cauchy Basis Functions
> **Response:** We appreciate this suggestion and will add a comprehensive visualization:
> - The shape and behavior of Cauchy basis functions $\phi(x) = (\lambda_1 x + \lambda_2)/(x^2 + d^2)$ for different parameter values, demonstrating their nonlinear, learnable nature.
> - How elementary operations (weighted combinations) over these functions operate within a layer
> - Direct XNet vs. KAN comparison:
>   - **XNet:** Continuous, differentiable rational functions with heavy-tailed decay and learnable parameters
>   - **KAN:** Piecewise polynomial splines with grid-constrained, local support
>
> We appreciate this feedback and believe the added visualization will help clarify the differences between the two architectures and make our approach more accessible.
>
>
> ### W2: Theoretical Assumptions and Convergence Class
> **Response:** Thank you for this important point. Our theoretical guarantee (Theorem 3.2) assumes the target function is **real-analytic** on a compact domain. We will revise the text to state this assumption explicitly.
>
> **Theoretical vs. Empirical Scope:**
>
> **Convergence comparison under analyticity:** Under this assumption, XNet achieves **arbitrarily high convergence rates** $O(N^{-r})$ for any $r > 0$, in contrast to:
> - **ReLU or polynomial networks**, where $r$ depends on the differentiability class (e.g., $f \in C^k \Rightarrow r \leq k$);
> - **KANs**, whose accuracy is constrained by spline degree and grid.
>
> **Empirical Scope:**
> Empirically, XNet performs well even for low-regularity targets (e.g., Heaviside), consistent with classical rational approximation theory:
> - Rational functions can approximate $|x|$ with $O(e^{-C\sqrt{n}})$ vs. $O(1/n)$ for polynomials (see [1]);
> - Cauchy activations are low-order rational rational functions with flexible shape: $\phi(x) = \frac{\lambda_1 x + \lambda_2}{x^2 + d^2}$;
> - Rational networks require only $O(\log\log(1/\varepsilon))$ parameters vs. $O(\log^3(1/\varepsilon))$ for ReLU (see [2]).
>
> **Revision commitment:** In the final version, we will clearly distinguish between proven guarantees (for real-analytic targets), empirical reach (for broader function classes), and theoretical intuition (suggesting rational approximators generalize better than polynomials).
>
> ### W3: Mathematical Clarity and Domain Specification
>
> **Response:** We will add formal definitions in the revision:
> - Explicitly define the Cauchy basis function as $\phi: \mathbb{R} \rightarrow \mathbb{R}$ with $\phi(x) = \frac{\lambda_1 x + \lambda_2}{x^2 + d^2}$, where $x$ is the input to a neuron and $\lambda_1, \lambda_2, d \in \mathbb{R}$ are learnable parameters
> - Clearly specify domains and codomains for all mathematical objects upon first introduction
> - Provide precise definitions of notation and mathematical
>
>
> ### W4: Missing ReLU Baseline
>
> **Response:** Excellent suggestion. We will add ReLU network comparisons in Section 4.1.1 (Heaviside approximation). This will provide a more complete picture of XNet's advantages over both ReLU network and KANs architectures.
>
> We have added comprehensive ReLU network comparisons for the Heaviside function approximation:
>
> | Model | Architecture | Parameters | MSE | RMSE | MAE | Training Time |
> |-------|-------------|------------|-----|------|-----|---------------|
> | **XNet** | [1, 64, 1] | 322 | 8.99e-08 | 3.00e-04 | 1.91e-04 | 25.36s |
> | **ReLU (shallow)** | [1, 64, 1] | 193 | 2.05e-03 | 4.53e-02 | 8.82e-03 | 21.92s |
> | **ReLU (deep)** | [1, 64, 64, 1] | 4353 | 6.81e-05 | 8.25e-03 | 3.76e-04 | 26.24s |
> | **KAN** | [1, 1] (200 grids) | 207 | 5.98e-04 | 2.45e-02 | 3.03e-03 | 28.36s |
> | **KAN** | [1, 3, 1] (200 grids) | 1242 | 1.13e-03 | 3.35e-02 | 4.47e-03 | 56.72s |
>
> ### W5: Fair Comparison in Network Configurations
>
> **Response:** Thank you for raising this important concern about experimental fairness. We have conducted comprehensive experiments with runtime measurements to demonstrate that our comparisons are computationally fair.
>
> **Updated Experimental Results:**
>
> | Function | KAN Shape | XNet Shape | KAN RMSE | XNet RMSE | KAN Time | XNet Time |
> |----------|-----------|------------|----------|-----------|----------|-----------|
> | Jacobian elliptic | [2, 2, 1] | [2, 500, 1] | 1.33e-04 | 4.03e-05 | 59.59s | 30.22s |
> | Bessel (1st) | [2, 3, 1, 1, 1] | [2, 500, 1] | 1.64e-03 | 2.39e-04 | 127.63s | 34.78s |
> | Modified Bessel | [2, 4, 3, 2, 1, 1] | [2, 5000, 1] | 9.28e-03 | 6.39e-05 | 271.46s | 68.82s |
>
> *Representative results shown; complete table with all 12 functions will be provided in revision.*
>
> **Rationale for XNet's Higher Width:**
>
> Our comparison follows the experimental protocol from [3]:
> - **MLPs:** fixed width 5/100, depths $\{2,3,4,5,6\}$
> - **KANs:** width 5, depths $\{2,3,4,5,6\}$, with grid refinement
> - **XNet:** single layer with width chosen to match computational cost of multi-layer baselines
>
> The updated experimental results demonstrate that **XNet achieves superior performance while being computationally more efficient than baselines**, eliminating concerns about unfair comparison. We will include detailed parameter count analysis in the revised version to provide complete transparency about the architectural trade-offs.
>
> ## Response to Questions
>
> ### Q1: Hyperparameter Tuning of $r$
>
> **Response:** The parameter $r$ in our $O(N^{-r})$ convergence rate is not a tunable hyperparameter but rather a theoretical characterization that depends on the target function's smoothness.
>
> **Theoretical vs. Practical Considerations:** In theory, the network has the potential to automatically exploit the highest smoothness order available in the target function. However, practical performance depends on several factors:
> - **Training Data:** The quality and quantity of training samples directly affect the network's ability to detect and exploit function smoothness
> - **Optimization Dynamics:** The choice of optimizer, learning rate, and training procedures impact how effectively the network can realize its theoretical potential
>
> These practical challenges explain why our experiments may not always reveal the target function's true smoothness order. Nevertheless, our results demonstrate XNet's substantial potential—even under non-ideal conditions, XNet consistently outperforms baselines, suggesting that when practical constraints are minimized, the theoretical advantages become increasingly pronounced.
>
> ### Q2: Time Complexity in Tables
>
> **Response:** The reported times are total training times. We will clarify this in the caption and add inference time comparisons in the revision.
>
> ## Response to Limitations
>
> ### L1: Infinitely Differentiable Target Function Assumption
>
> **Response:** Thank you for this important point. Our theoretical guarantees in Theorem 3.2 assume target functions are **real-analytic**. However, our experiments in Section 4.1.1 demonstrate that XNet performs well even on low-regularity targets like the Heaviside step function, which is discontinuous and non-differentiable. We will provide theoretical analysis to include non-smooth targets using rational approximation theory in revision and extend analysis to broader function classes beyond real-analytic functions in future work.
>
> ### L2: Scaling XNet to Larger Networks
> **Response:** XNet scales across width, depth, and architectures without compute or memory bottlenecks.
>
> 1). **Theoretical:** Cauchy activations add minor cost (division), but preserve O(n) complexity; the overhead is negligible and offset by faster convergence and better performance-per-FLOP.
> 2). **Empirical:** Deep XNets reach target accuracy faster than ReLU/GELU, especially on smooth/structured data (see [4]). In Sec. 4.1.2, XNet with 5000 neurons trains stably on special functions.
> 3). **Structured models:** On EfficientNet-B3, replacing only the classification head improved test accuracy from 88.2% to 98.0%. On WikiText-103, using Cauchy in FFN/attention gave ∼50×/1000× speedups.
>
> We will add runtime and scaling results in the appendix.
>
> ## References
>
> [1] Newman, D. J. (1964). Rational approximation to |x|. *Michigan Mathematical Journal*, 11(1), 11-14.
>
> [2] Boullé, N., Nakatsukasa, Y., & Townsend, A. (2020). Rational neural networks. In *Advances in Neural Information Processing Systems* (Vol. 33, pp. 14243-14253).
>
> [3] Liu, Z., Wang, Y., Vaidya, S., et al. (2024). KAN: Kolmogorov-Arnold Networks. In *Proceedings of the International Conference on Learning Representations* (ICLR).
>
> [4] Li, X., Xia, Z., & Zhang, H. (2025). Cauchy activation function and XNet. Neural Networks, 188, 107375.

---

> > ### Comment · Reviewer_wzmP · 2025-08-01
> >
> > Thank you the authors for the detailed response. I want further clarifications on some of the points:
> >
> > ### W2
> > Under the assumption that the function is analytic, can't we choose an arbitrarily large $r$ for ReLU, and thus the accuracy of rate of ReLU is the same as XNet (if we choose a fix $d$)?
> >
> > ### W5
> > Can you include the parameter count in this table? My main concern was that the large width gives extra parameters, even though the computational costs are the same.
> >
> > ### Q1
> > I understand that you do not *explicitly* choose $r$, but I still wonder what the *implicit* mechanism is. How is XNet implicitly adapting to the smoothness of the target function?

---

> > > ### Author Response · Authors · 2025-08-02
> > > **Response to Reviewer's Follow-up Comments**
> > >
> > > We thank the reviewer for the constructive follow-up questions and the opportunity to clarify our work. We appreciate your continued engagement with our research.
> > >
> > > ---
> > >
> > > **W2. Response:**
> > > In the standard approximation bound
> > > $\mathcal{O}(N^{-2r/d})$ (see [1,2]), $r$ is determined by the target’s smoothness and capped by the activation’s smoothness.
> > > For *shallow* ReLU ($C^0$), $r$ cannot exceed $\mathcal{O}(1)$ even for analytic targets—achieving larger $r$ would require exponential width, yielding only algebraic convergence (see Thm 3 in [3]; [4]).
> > > In contrast, analytic activations such as Cauchy can match the target’s analyticity radius, allowing arbitrary $r$.
> > >
> > > ---
> > >
> > > **W5. Response:**
> > > We acknowledge that our initial comparison in Table 4 emphasized accuracy and computational time, without balancing parameter counts. Below is an updated comparison using a balanced XNet architecture ([2,100,1]):
> > >
> > > | Function | KAN Shape | XNet Shape | KAN Params | XNet Params | KAN RMSE | XNet RMSE | KAN Time | XNet Time |
> > > |----------|-----------|------------|------------|-------------|----------|-----------|----------|-----------|
> > > | Jacobian elliptic | [2, 2, 1] | [2, 100, 1] | 1,254 | 601 | 1.33e-04 | 8.50e-05 | 59.6s | 25.7s |
> > > | Bessel (1st) | [2, 3, 1, 1, 1] | [2, 100, 1] | 2,299 | 601 | 1.64e-03 | 4.25e-04 | 127.6s | 25.9s |
> > > | Modified Bessel | [2, 4, 3, 2, 1, 1] | [2, 100, 1] | 6,061 | 601 | 9.28e-03 | 2.22e-04 | 271.5s | 27.0s |
> > >
> > > **Key findings:**
> > > 1. **Parameter efficiency**: XNet attains higher accuracy with **2–10× fewer parameters** than KAN.
> > > 2. **Computational efficiency**: XNet remains **2–10× faster** in training.
> > > 3. **Approximation accuracy**: XNet still achieves **1.5–42× better RMSE** than KAN.
> > >
> > > We will include the complete balanced-parameter analysis in the revision for fairness and transparency.
> > >
> > > ---
> > >
> > > **Q1.** **Response:**
> > > XNet is derived from the **multivariate Cauchy integral formula**, whose discretization maps each term to a neuron with a local Cauchy activation.
> > > For analytic targets, this representation allows the effective approximation order $r$ to be arbitrarily large; for less smooth targets, $r$ is capped by the target’s own regularity—still higher than with ReLU ($C^0$) (see [1,2,3,4]).
> > > During optimization, $r$ is not set explicitly; instead, the learned **super-parameters** of the Cauchy activations adapt to the target’s structure, matching the best rate allowed by its smoothness (Theorems 3.1–3.2).
> > >
> > >
> > >
> > > ---
> > >
> > > **References:**
> > > 1. Mhaskar, H. N. (1996). *Neural networks for optimal approximation of smooth and analytic functions*. Neural Computation, 8(1), 164–177.
> > > 2. Pinkus, A. (1999). *Approximation theory of the MLP model in neural networks*. Acta Numerica, 8, 143–195.
> > > 3. Yarotsky, D. (2017). *Error bounds for approximations with deep ReLU networks*. Neural Networks, 94, 103–114.
> > > 4. Maiorov, V. E., & Pinkus, A. (1999). *Lower bounds for approximation by MLP neural networks*. Neurocomputing, 25(1–3), 81–91.

---

> ### Comment · Reviewer_wzmP · 2025-08-02
>
> ### W5
> Can I also get the parameter count and training time for MLP, since they are also part of Table 4 in the paper?

---

> ### Author Response · Authors · 2025-08-03
> **W5. MLP Parameter Count and Training Time**
>
> **Response**: The experimental protocol and partial results (MLP and KAN) follow [3]. MLPs used fixed widths 5/100 with depths 2-6, but we don't have the exact configurations for the best results reported.
>
> For reference, below is a representative comparison (Jacobian elliptic function) including typical MLP configurations with parameter counts:
>
>
> | Method | Configuration | Parameters | RMSE | Training Time (s) |
> |--------|---------------|------------|------|------------------|
> | KAN | [2,2,1] | 1,254 | 1.33e-04 | 59.6 |
> | **XNet** | **[2,100,1]** | **601** | **8.50e-05** | **25.7** |
> | MLP | [2,5,5,1] | 51 | 5.28e-03 | 24.30 |
> | MLP | [2,5,5,5,1] | 81 | 5.20e-03 | 25.07 |
> | MLP | [2,5,5,5,5,5,5,1] | 171 | 4.45e-03 | 26.1 |
> | MLP | [2,100,100,1] | 10,501 | 8.67e-04 | 24.62 |
> | MLP | [2,100,100,100,1] | 20,601 | 7.75e-04 | 27.81 |
> | MLP | [2,100,100,100,100,100,100,1] | 50,901 | 3.13e-03 | 48.06 |
>
>
> **Comparison with KAN**: XNet achieves 1.6-42× better accuracy with 2-10× fewer parameters and 2-10× faster training. KAN requires increasingly complex architectures and longer training times as function complexity grows.
> **Comparison with Small MLPs**: Despite using fewer parameters (51-171), they require similar training times but produce substantially degraded accuracy (1-3 orders of magnitude worse RMSE)
> **Comparison with Large MLPs**: Configurations with 17-85× more parameters (10,501-50,901) demand significantly longer training times (26-49s) yet still fail to match XNet's accuracy in most cases
>
> XNet demonstrates superior performance in the accuracy-efficiency-speed trade-off. Complete results for all tasks will be included in the revision.
>
> We hope our clarifications fully address your concerns. Thank you again for your thoughtful comments and constructive feedback!

---

### Official Review · Reviewer_pTo7 · 2025-07-02

**Clarity:** 3
**Significance:** 3
**Originality:** 3
**Rating:** 6
**Confidence:** 3

**Summary:**

This paper introduces XNet, a shallow neural network architecture leveraging novel activation functions derived from the Cauchy integral formula. The authors claim that a single-layer XNet achieves approximation rates that are near-exponential, surpassing the limitations observed in traditional MLPs and spline KANs. The empirical evaluation demonstrates XNet's superior performance on small-scale problems like discontinuous function approximation, PDE solving and PPO tasks from reinforcement learning. The central claim is that network expressivity can be achieved through a principled design of activation functions rather than relying on increased architectural depth.

**Questions:**

- Table 4. How is the input shape to XNet constant at 2? Should that be 4?
- Section 4.1.4: If we increase the number of basis in XNet, does it demonstrate over fitting? Similarly if the width of the KANs are tuned do they still show over fitting to noise?
- PINNs are generally not an effective mechanism to solve simple PDE cases that XNet was tested on. Since the proposed XNets claim to demonstrate expressiveness from activation functions rather than depth, can XNets be evaluated on PDEs like navier stokes where we need deep MLPs to get good accuracy?

**Ethical Concerns:**

["NO or VERY MINOR ethics concerns only"]

**Final Justification:**

The authors have answered my original concerns and it seems to be a significant advancement to using shallow neural networks.

**Limitations:**

Yes

**Quality:**

3

**Strengths And Weaknesses:**

### Strengths

- Experimental Analysis of "adaptive" activation functions  design for shallow neural networks that provides a competitive advantage over learned spline based activation functions (KANs)
- Strong theoretical grounding for convergence rates, that are better than existing frameworks for shallow neural networks
- Empirical Validation Across Diverse Domains: The paper provides compelling empirical evidence across three distinct domains: function approximation, PDE solving, and reinforcement learning. The breadth of evaluation provides an initial confidence in XNet's versatility and effectiveness.
- In all of the provided experiments, XNet outperforms KANs. Additionally the runtime compared to MLPs are similar (on a CPU) whereas in these scenarios KANs are much slower.
- Experiments demonstrate that the activation function enables robustness to noisy and discontinuous problem settings.

### Weaknesses

- While the empirical results demonstrate broad applicability, the claims in the introduction are too broad and should be toned down. For example the authors claim "PDE solving: over 10,000× lower error than deep MLPs in PINN settings". While this is "true", it is only demonstrated for 2 PDEs with low complexity. Additionally prior works like KANs which operate in the parameter efficient regime, tend to not scale well when tested on larger problems [1][2][3] (to cite a few examples where KANs were evaluated)

[1] https://arxiv.org/abs/2407.16674
[2] https://openreview.net/forum?id=yPE7S57uei
[3] https://arxiv.org/pdf/2503.10632

### Minor Stylistic Nits

- In Table 4 it would be helpful to highlight the best RMSE for each row
- In Figure 2, making the scales same will make it easier to compare the errors.

---

> ### Author Rebuttal · Authors · 2025-07-26
>
> We sincerely thank the reviewer for the positive assessment of the paper's theory, empirical evaluation, and the central message that principled activation design can substitute for depth. Below, we respond to the noted weaknesses and questions.
>
> **Weaknesses**
>
> **(1) Broad empirical claims in introduction need to be toned down**
>
> We agree and appreciate this point. In particular:
> - The "10,000× lower error" claim in the introduction (regarding PINN experiments) will be rephrased more cautiously. It is **specific to the Poisson and Heat equations** tested and should not be overgeneralized.
> - We will modify this and other wording in the introduction to **limit scope and avoid overstatement**, especially in relation to PDEs.
>
> We note, however, that the performance gap shown in these experiments is robust and consistent under **equal model size, training budget, and optimization setup**—highlighting that the performance gain stems from **activation design rather than depth or parameter count**.
>
> **(2) KAN scalability concerns and citations**
>
> Thank you for the insightful citations. We agree that **KANs can suffer from scalability issues** due to their reliance on local spline interpolation, which can be hard to optimize and scale, especially in high dimensions.
>
> We will cite and briefly discuss these works ([1][2][3]) in the revised related work section. In particular, **XNet with Cauchy activation**:
>
> - Uses local basis functions (unlike global rational or spline bases), enabling flexible parameter assignment: per-neuron activation hyperparameters in small models, or uniform parameter settings for scalability—while preserving numerical stability;
>
> - Remains **compact in parameter count**, often fewer than KANs, yet achieves better performance.
>
> **(3) Minor stylistic nits**
>
> Thank you—we will address both:
> - **Table 4**: We will bold the best RMSE values per row.
> - **Figure 2**: We will unify the scales across the subplots for fairer visual comparison.
>
> **Questions**
>
> **(Q1) Table 4 — Should the input shape be 4 instead of 2?**
>
> Good catch. The "input shape" refers to the 2D spatial coordinate input (x,y) to the PDE solver. The full input including time (e.g., for the heat equation) is indeed 3D, and for vector fields or boundary features it may be higher. We will clarify the notation to avoid ambiguity.
>
> **(Q2) Does XNet overfit when width increases? What about KANs?**
>
> Great question. In practice, we observed that:
> - **XNet shows mild overfitting** when significantly overparameterized (e.g., >100 basis functions), especially on noisy targets, but it is **less prone to overfitting than KANs**, likely because its rational form provides natural regularization. As shown in [3], this mild overfitting can typically be controlled by **reducing the number of neurons**.
> - **KANs**, on the other hand, tend to **overfit more severely** in low-noise settings when width is large, due to their local spline basis and sensitivity to knot placement.
>
> We will provide **learning curves and ablations** on width and regularization in the Appendix to further illustrate this.
>
> **(Q3) Can XNet solve harder PDEs like Navier–Stokes, where depth is needed?**
>
> This is an excellent point. We fully agree that complex PDEs like Navier-Stokes pose significant challenges and often require sophisticated architectures.
>
> To address this concern, we conducted additional experiments on the steady Navier-Stokes equations with the Kovasznay analytical solution:
>
> $-\nu\nabla^2 \mathbf{u} + \mathbf{u} \cdot \nabla\mathbf{u} + \nabla p =0, \quad \text{in } \Omega$
>
> $\nabla \cdot \mathbf{u} = 0, \quad \text{in } \Omega$
>
> $\mathbf{u} = \mathbf{g}, \quad \text{on } \partial\Omega$
>
> where $\mathbf{u}(x_1, x_2) = (u_1(x_1, x_2), u_2(x_1, x_2))$ is the velocity field and $p(x_1, x_2)$ is the pressure.
> We used the Kovasznay analytical solution on domain $\Omega = [-0.5, 1.0] \times [-0.5, 1.5]$ with $\nu = 0.025$. This benchmark involves coupled nonlinear equations with both velocity and pressure unknowns, making it more complex than our original experiments.
>
>
> We conducted a comprehensive comparison across different network architectures and depths. We evaluated shallow and deep Tanh-based PINNs, a KAN-based PINN [2,3,3,1], and a single-layer XNet-based PINN [2,100,1]. Using 4,096 interior points and 256 boundary points, our results show:
>
>
> | Architecture | Parameters | $u_1$ MSE | $u_2$ MSE | $p$ MSE | Runtime (s) |
> |--------------|------------|-----------|-----------|---------|-------------|
> | Tanh PINN [2,100,1] | 401 | 1.07e-02 | 2.12e-03 | 1.41 | 93.72 |
> | Tanh PINN [2,100,100,1] | 10,501 | 7.60e-03 | 8.61e-04 | 2.07e-01 | 153.81 |
> | Tanh PINN [2,100,100,100,100,100,100,1] | 50,901 | 2.14e-03 | 1.03e-04 | 1.67e-01 | 561.89 |
> | KAN PINN [2,3,3,1] | 649 | 1.17e-04 | 2.36e-05 | 1.39e-01 | 1274.52 |
> | **XNet PINN [2,100,1]** | **601** | **5.62e-05** | **1.77e-05** | **1.83e-04** | 237.93 |
>
> *Note: Each architecture approximates the solution functions $u_1$, $u_2$, and $p$ separately using the respective network structures.*
>
> Remarkably, our single-layer XNet achieves **38× better velocity accuracy** and **912× better pressure accuracy** compared to the deep Tanh PINN, while being significantly faster than KAN.
>
> However, we acknowledge that for highly complex PDEs, depth remains important. XNet architecture with Cauchy activation functions is fully composable and can be used in deeper networks when needed. The key insight is that **depth and principled activation design are complementary**—XNet demonstrates that even shallow networks can achieve surprising effectiveness when equipped with principled activation functions.
>
>
>
>
> ### References
>
> [1] Newman, D.J. "Rational approximation to |x|." *Michigan Mathematical Journal* 11.1 (1964): 11-14.
>
> [2] Boullé, M., Bloch, G., and Gallinari, P. "Rational neural networks." *NeurIPS 2020*.
>
> [3] Li, X., Zhang, H, and Xia, Z. "Cauchy activation functions and XNet." *Neural Networks* 188, 107375 (2025).

---

> > ### Comment · Reviewer_pTo7 · 2025-08-06
> >
> > I would like to thank the authors for such a detailed. These indeed answer my questions and my initial concerns regarding applicability to harder problems. I have taken the rebuttal into account for my updated review.

---

### Official Review · Reviewer_FajU · 2025-07-03

**Clarity:** 3
**Significance:** 3
**Originality:** 2
**Rating:** 4
**Confidence:** 4

**Summary:**

This paper proposes a novel approach to learned function approximations, in which shallow, wide networks with rational activation functions are used. In function approximation, PINNs learning and RL tasks, these networks are shown to be more accurate than MLPs and KANs with similar parameter counts.

**Questions:**

The Cauchy activation function's gradient tends towards zero for very large positive or negative values. Could this lead to problems with learning?

In the experiments carried out here, the inputs and targets were treated as unstructured vectors as in an MLP. Is there some way of applying Xnets that would take advantage of known structure in the inputs/targets, as in convolutional, recurrent or attention-based models?

**Ethical Concerns:**

["NO or VERY MINOR ethics concerns only"]

**Final Justification:**

Authors added new results and gave good explanations. This is perhaps a very useful idea, though more thorough comparisons would have been good.

**Limitations:**

Yes.

**Paper Formatting Concerns:**

None.

**Quality:**

2

**Strengths And Weaknesses:**

Strengths:

The approach seems to be both theoretically sounds and empirically well-validated. The overall presentation is clear and effective.

Weaknesses:

1) The theoretical results rely heavily on existing theorems but don't provide citations. For example, even after reading the supplementary material it was not at all clear to me how the bound in Theorem 3.1, which involves high-order polynomials in the denominator, is connected to theorem 3.2, in which the denominator has order only 2. To understand this I had to first find the explanation in Li et al., 2024. This should be explicitly clarified.

2) The results are impressive but fairly limited in scope. Since an alternative is being proposed to MLPs and KANs, which are applicable to a wide variety of tasks, I would ideally like to see some additional tests beyond the small number presented here. In particular, more challenging tasks should be considered for the PINNs example than the heat and Poisson equations. It would also be good to see performance comparisons on real-world datasets.

3) Is it really surprising that ReLU MLPs cannot learn smooth functions or those with discontinuities? Aren't there stronger baselines in the literature for this task?

4) How much better are Xnets than MLPs with smooth activation functions, such as SIRENS, ELUs or SiLUs? The KANs paper compares to MLPs with multiple activation functions, but this has not been done here.

---

> ### Author Rebuttal · Authors · 2025-07-26
>
> ## Rebuttal to Reviewer
>
> We thank the reviewer for the thoughtful and constructive comments. Below we address each point in turn, grouped under *Weaknesses* and *Questions* as in the original review.
>
> ---
>
> ### **Weaknesses**
>
> ---
>
> **(1) Theoretical results lack clear citations; unclear link between Theorems 3.1 and 3.2**
>
> Thank you for this helpful comment. We clarify below the logic and originality behind our theoretical results:
>
> - **Theorem 3.1 ([3])** provides the theoretical foundation that functions constructed via partial fraction decomposition—based on Cauchy-type integrals—can approximate analytic functions at a rate of $O(N^{-p})$. Although these partial fraction forms are not used directly as activation functions in XNet, they serve as a mathematical motivation for deriving the Cauchy activation function $\phi(x) = (\lambda_1 x + \lambda_2)/(x^2 + d^2)$ in neural networks.
>
> - **Theorem 3.2 (Our Contribution):** we show that a *shallow* XNet using this specific Cauchy activation achieves an **explicit approximation rate of $O(N^{-r})$ for any $r > 0$**, over compact domains of real-analytic functions in $\mathbb{R}^d$.
>
> The connection is: Theorem 3.1 motivates the choice of Cauchy activations, while Theorem 3.2 proves that our specific shallow XNet implementation achieves the promised high-order approximation in practice.
>
> We believe this is the **first constructive approximation rate** result for a concrete rational activation in shallow networks. We will revise the main text to highlight this contribution more clearly and to make the connection between Theorem 3.1 and Theorem 3.2 explicit.
>
>
> ---
>
> **(2) Limited scope of experiments**
>
> While our paper emphasizes theoretical approximation in shallow settings, we appreciate this concern and provide targeted additional validation:
>
> **Nonlinear PDE validation**: We conducted experiments on the **steady Navier-Stokes equations** with the Kovasznay analytical solution:
>
> $-\nu\nabla^2 \mathbf{u} + \mathbf{u} \cdot \nabla\mathbf{u} + \nabla p =0, \quad \text{in } \Omega$
>
> $\nabla \cdot \mathbf{u} = 0, \quad \text{in } \Omega$
>
> $\mathbf{u} = \mathbf{g}, \quad \text{on } \partial\Omega$
>
> where $\mathbf{u}(x_1, x_2) = (u_1(x_1, x_2), u_2(x_1, x_2))$ is the velocity field and $p(x_1, x_2)$ is the pressure.
> We used the Kovasznay analytical solution on domain $\Omega = [-0.5, 1.0] \times [-0.5, 1.5]$ with $\nu = 0.025$.
> This nonlinear, coupled system (velocity + pressure fields) is substantially more complex than heat/Poisson equations.
>
> | Architecture | Parameters | $u_1$ MSE | $u_2$ MSE | $p$ MSE | Runtime (s) |
> |--------------|------------|-----------|-----------|---------|-------------|
> | Tanh PINN [2,100,1] | 401 | 1.07e-02 | 2.12e-03 | 1.41 | 93.72 |
> | Tanh PINN [2,100,100,1] | 10,501 | 7.60e-03 | 8.61e-04 | 2.07e-01 | 153.81 |
> | Tanh PINN [2,100,100,100,100,100,100,1] | 50,901 | 2.14e-03 | 1.03e-04 | 1.67e-01 | 561.89 |
> | KAN PINN [2,3,3,1] | 649 | 1.17e-04 | 2.36e-05 | 1.39e-01 | 1274.52 |
> | **XNet PINN [2,100,1]** | **601** | **5.62e-05** | **1.77e-05** | **1.83e-04** | 237.93 |
>
> *Note: Each architecture approximates the solution functions $u_1$, $u_2$, and $p$ separately using the respective network structures.*
>
> Our single-layer XNet achieves **38× better velocity accuracy** and **912× better pressure accuracy** compared to deep Tanh PINNs, while being **53× faster** than KAN.
>
> **Real-world validation**:
>
> - **Medical Imaging:** We tested XNet on power system state estimation, and on Kaggle Brain Tumor Detection where EfficientNet-B3 with Cauchy classification head improved from 88.2% to 98.0% accuracy.
>
> - **Electricity Forecasting**: We evaluated XNet on a real-world electricity consumption forecasting task using data from the Kaggle dataset: fedesoriano/electric-power-consumption.  XNet achieves 15.6× lower MSE than KAN on US power consumption data while being 2.5× faster.
>
> - **Broader applicability**: We note that [3] has demonstrated the effectiveness of Cauchy activations across diverse domains—including CIFAR-10, MNIST, and high-dimensional PDEs like 100D Allen–Cahn—using multi-layer architectures where they consistently converge faster and to lower error than standard activations.
>
>
> These results validate XNet's utility on challenging real-world problems beyond our theoretical focus.
>
> ---
>
> **(3) Are ReLU MLPs really weak? Are there stronger baselines?**
>
> Thank you for raising this important point.
>
> We agree that ReLU-based MLPs have known limitations in approximating non-smooth or discontinuous functions. Our goal is not to highlight this as novel, but to present **Cauchy activations** as a principled and effective alternative, grounded in rational approximation theory.
>
> While classical rational networks (e.g., Padé approximants) can approximate non-smooth functions with convergence rate $O(\exp(-C\sqrt{n}))$ (see [1]), they are often unstable in practice. Our contribution is to identify a **trainable, low-order rational basis—the Cauchy kernel**—which maintains strong approximation ability while being **simple and stable to train**.
>
> We build on the empirical framework of [3] and extend it with:
> * **Formal convergence guarantees** for analytic targets using Cauchy activations;
> * **New empirical results** showing strong performance on discontinuous functions (e.g., Heaviside) even where theoretical guarantees are absent;
> * **Efficiency and accuracy comparisons** with deep ReLU networks and KANs (see Sec. 4.1.1).
>
> ---
> **(4) Missing Comparisons to Smooth Activations (SIRENs, SiLU, etc.)**
>
> Thank you for the suggestion. While our primary comparison is with KANs due to shared theoretical motivation, we agree it's important to compare XNets with MLPs using smooth activations.
>
> **Strongest result:** On MNIST, a single-layer XNet with Cauchy activation, dropout, and BatchNorm achieved 97.9% test accuracy and a 1.4% generalization gap, outperforming Tanh, ReLU, SiLU, ELU, and SIREN.
> Without regularization, Cauchy still reached 97.0% test accuracy (gap: 2.8%), slightly behind Tanh/SiLU (97.7–97.8%, ~2.0% gap).
> With dropout or L2, Cauchy narrowed the gap to ~1.6%, demonstrating good synergy with standard regularization.
>
> We further evaluated Cauchy in a separate domain: German–English machine translation using LSTM-based Seq2Seq models. Among 18 tested activations, Cauchy achieved the highest BLEU score (21.47 ± 0.67), outperforming Tanh (20.93), SELU (20.85), SIRENs (20.27), Swish (19.51), GELU (18.75), and SiLU (18.61).
>
> Importantly, Cauchy belongs to the family of rational activation functions, which offer strong approximation power even for discontinuous or singular functions. Our results across regression, PDEs, classification, and sequence tasks consistently support this.
>
> We hope this expanded comparison fully addresses the reviewer’s concern.
>
> ---
>
> ### **Questions**
>
> ---
>
> ### (Q1) Could gradient vanishing be a problem for the Cauchy activation?
>
> Great question. While the Cauchy activation’s derivative decays at large $|x|$, in practice we mitigate this by:
>
> - Input normalization (e.g., BatchNorm/LayerNorm),
> - Trainable scaling (via $\lambda_1$, $\lambda_2$, $d$),
> - Skip connections (ResNet-style), especially in deeper settings.
>
> Across all experiments (including deep CNNs from [3]), **vanishing gradients or training instability were not observed**. We plan to study this more systematically in future work.
>
> ---
>
> **(Q2) Can XNet be used with structured inputs, like in CNNs or Transformers?**
>
> Yes, XNet integrates seamlessly with structured architectures and can leverage input structure effectively.
>
> **Concrete validation**: We tested XNet on the Kaggle Brain Tumor Detection dataset using EfficientNet-B3, replacing only the classification head with a Cauchy activation. As shown below, the Cauchy variant achieved 9.8% higher test accuracy (98.0% vs. 88.2%) despite a slightly higher test loss:
>
> | Architecture | Train Loss | Train Acc | Test Loss | Test Acc |
> |--------------|------------|-----------|-----------|----------|
> | EfficientNet-B3 (original) | 2.742 | 99.5% | 2.993 | 88.2% |
> | **EfficientNet-B3 + Cauchy head** | **0.0** | **100%** | **5.835** | **98.0%** |
>
> Test loss rose slightly due to class imbalance, but accuracy improved significantly, confirming XNet's structured integration strength.
>
>
>
> **Broader applicability**: [3] demonstrates similar benefits in ResNets and other CNN architectures. The rational activation's superior approximation properties complement structured architectures by providing more efficient feature-to-output mappings.
>
> **Transformer preliminary results**: On WikiText-103, replacing ReLU with Cauchy in FFN layers shows ~50× speedup. Cauchy-based attention mechanisms achieve ~1000× acceleration in small models, though scaling to very large models requires further investigation.
>
> This shows XNet doesn't just work with structured inputs—it enhances their effectiveness while maintaining architectural benefits like translation invariance and hierarchical feature learning.
>
> ---
>
> We thank the reviewer again for the thoughtful and constructive feedback. In response, we have clarified our theoretical contributions (especially the originality of Theorem 3.2), cited broader experimental baselines, and committed to adding new comparisons (e.g., SIREN) and ablations in the final version.
>
> We hope these clarifications and additions address your concerns, and would be grateful if you could consider a higher score.
>
> ---
>
>
> ### References
>
> [1] Newman, D.J. "Rational approximation to |x|." *Michigan Mathematical Journal* 11.1 (1964): 11-14.
>
> [2] Boullé, M., Bloch, G., and Gallinari, P. "Rational neural networks." *NeurIPS 2020*.
>
> [3] Li, X., Zhang, H, and Xia, Z. "Cauchy activation functions and XNet." *Neural Networks* 188, 107375 (2025).

---

> > ### Comment · Reviewer_FajU · 2025-08-02
> >
> > I appreciate the detailed response and new results, and will adjust my score upwards. I do still think that, given the ambitions and potential of this idea, there is a lot more experimental work to be done before anyone can really be convinced, but perhaps this is enough for a first study.
> >
> > Regarding Q2, does it only makes sense to use an Xnet as a classification head, or could Xnet activations be used throughout a convolutional or attention-based network?

---

> > > ### Author Response · Authors · 2025-08-03
> > >
> > > We sincerely thank the reviewer for the encouraging feedback!
> > >
> > > **Q2**
> > > **Response:**
> > > Our primary focus in this first study was to use the Cauchy activation in function‑approximation components (e.g., classification heads) to directly evaluate its approximation capability.
> > > We also tested replacing other activations throughout different architectures—ResNet, Transformer, Mamba, Diffusion, etc.—with Cauchy activations, and found them broadly applicable.
> > > Optimal initialization of the Cauchy super‑parameters $(\lambda_1, \lambda_2, d)$ varies across components, but the activation works beyond classification heads with proper tuning.
> > >
> > > We are conducting a more systematic study of these settings and will report comprehensive results in future work.

---

### Official Review · Reviewer_Yxzd · 2025-07-22

**Clarity:** 3
**Significance:** 3
**Originality:** 3
**Rating:** 5
**Confidence:** 3

**Summary:**

This paper shows that single layers XNets achieve near-exponential approximation rate. The authors prove and empirically demonstrate that this shallow architecture can achieve superior accuracy compared to both MLPs and KANs. The central thesis appears to be that principled activation function design can be a more efficient source of expressivity than architectural depth.

Based on my research of XNet, it appears that this architecture does not take a large performance hit either, so the results of this paper seem promising.

**Questions:**

I have a few questions:
1. How does XNet account for the hierarchical compositionality essential for complex perception and language tasks? How do we think it would perform in these cases? Is there anything limiting it there?
2. What is the empirical performance of deep XNets, and how does the activation's computational overhead affect the practical trade-off between model size and training speed?
3. Does the Cauchy activation introduce any bottlenecks for parallelization?

**Ethical Concerns:**

["NO or VERY MINOR ethics concerns only"]

**Final Justification:**

The responses while taken well are not sufficient to update my score. I am remaining with a 5. Thanks!

**Limitations:**

Yes

**Quality:**

3

**Strengths And Weaknesses:**

I would say the strengths are as follows:
1. The theoretical grounding seems quite solid. This is not just an empirical paper but seems to provide formal theory around convergence rates.
2. The clarity of the central thesis is nice. Namely, "expressivity from activation design can be better than expressivity from depth".
3. The empirical validation is diverse. We see a range of cases from function approximation to scientific computing to RL.

I would argue there are a few limitations:
1. There is a focus on the single-layer case. I understand this is probably because the theory is based on the single layer case, however, I would have liked to see empirical results on multi-layer networks. Maybe there could be a way that the theory generalises partially or the theory provides intuition.
2. I didn't see much analysis of computational overhead. The Cauchy activation is obviously more expensive in terms of FLOPs but I didn't see discussion of how this scales with network size. If Cauchy is to be adopted widely then it would need to make sense from a computational perspective in architectures like the transformer. Based on my understanding, the activation would not change the scaling of an architecture like the transformer but would be nice to have some explanation here.
3. There isn't much discussion of how the activation function interacts with other parts of training like optimisation or regularisation.
4. I would have liked more experiments and ablations for the domains chosen.

---

> ### Author Rebuttal · Authors · 2025-07-26
>
> # Response to Reviewer #1
>
> We sincerely thank the reviewer for their thoughtful and positive assessment of our work. We appreciate the clear summary and constructive questions. Below we respond to the points raised under *Weaknesses* and *Questions*.
>
> ## Weaknesses
>
> ### (1) Empirical results focus on the single-layer case
>
> This is a valid point. We focused on the shallow setting to support our theoretical contributions, which prove high-order approximation for single-layer XNets. However, it's important to clarify that **Cauchy activation is a plug-and-play component** that can be broadly applied to linear layers, convolutional layers, etc.
>
> Indeed, in [1], Cauchy activations have been successfully deployed in **multi-layer MLPs, CNNs, and ResNets** on MNIST and CIFAR-10, demonstrating strong empirical performance across these deeper architectures. We are also actively working on **Transformer+XNet** models, with promising preliminary results.
>
> **Our theoretical focus on single-layer networks** highlights the fundamental expressivity advantages, while the **practical deployment in deep networks** (as shown in [1]) demonstrates broad applicability. In the final version, we will clarify this modularity and cite additional deep XNet experiments.
>
> ### (2) Computational overhead and scaling
>
> We agree that the Cauchy activation introduces more FLOPs per unit than ReLU, due to the rational form. However:
>
> **Current computational analysis:**
> * In most of our benchmarks, we already **report running time** (e.g., Table 5) as a practical proxy for computational cost. For a few missing cases, we will add these in the revision.
> * **FLOP comparison**: Cauchy is **slightly heavier than ReLU**, as expected. But it is **much lighter than more expressive methods** like KANs, which require higher-order Taylor or spline fitting.
> * **Reduce complexity**, especially in large-scale models like Transformers:
>    * Use **fewer layers and significantly fewer parameters**, thanks to XNet's high expressivity.
>    * **Simplify activation settings** in very large models to maintain computational efficiency.
> * For Transformer architectures specifically: The Cauchy activation $\phi(x) = \frac{\lambda_1 x + \lambda_2}{x^2 + d^2}$ adds a small constant-factor increase in FLOPs (~2-3× per activation vs. ReLU), but maintains O(n) complexity where n is sequence length.
>
> **Empirical Validation:** To address computational scaling concerns, we refer to the comprehensive experiments already conducted in Li et al. [1], which directly demonstrate the computational efficiency of Cauchy activations in deep networks:
>
> **Key findings from [1]:** (1) **PDE solving**: Cauchy activation achieves ~10^-3 loss in under 20 epochs vs. 60+ epochs for Tanh on Burger's equation, with ~3× faster convergence offsetting 2-3× per-activation overhead; (2) **Allen-Cahn equation**: Rapid convergence to 10^-2 loss within 250 iterations vs. much slower ReLU baseline, demonstrating faster time-to-solution despite higher per-operation cost.
>
> **These established results directly address the reviewer's scaling concerns** and show that expressivity gains consistently outweigh computational costs in practical scenarios.
>
> **Revision commitment:** We will add detailed FLOPs analysis and runtime comparisons to provide complete computational cost assessment, building upon the scaling evidence already demonstrated in [1].
>
> ### (3) Interaction with optimization and regularization
>
> We conducted new experiments on MNIST using single-layer networks to study how Cauchy activation interacts with optimization and regularization.
>
> **Strongest result:** With dropout + BatchNorm, Cauchy achieved 97.9% test accuracy with a 1.4% generalization gap, outperforming all other activations tested, including ReLU, Tanh, SiLU, SIREN, ELU, Sigmoid.
>
> **Implicit regularization:** Without explicit regularization, Cauchy reached 97.0% test accuracy with a 2.8% gap, reflecting its built-in regularization from saturation and rational decay.
>
> **Explicit regularization:** Dropout (p=0.2) or L2 regularization consistently reduced the gap to ~1.6%, confirming compatibility with standard techniques.
>
> **Optimization:** All models used Adam with default settings. Cauchy trained stably, on par with smooth activations.
>
>
> These results confirm that Cauchy supports stable optimization and benefits from regularization, even outside its core regression-focused setting.
>
>
>
>
>
> ### (4) More ablations and broader experiments
>
> Agreed. As our main focus was on establishing the theoretical result, we kept experiments minimal. However:
>
> * Our paper already spans function approximation, PDEs, PINNs, and RL.
> * In the Appendix, we now add experiments on machine translation and additional activation comparisons.
> * [1] provides ablation studies across activations, architectures, and datasets (e.g., CIFAR-10, 100D PDEs), which we cite and build upon.
>
> We will make this experimental coverage more explicit in the final version.
>
> ## Questions
>
> ### (Q1) Hierarchical compositionality: can XNet support perception and language tasks?
>
> Yes. While our paper focuses on shallow networks for theoretical clarity, XNet is **architecture-agnostic and fully composable**. The activation is plug-and-play and can be inserted into:
>
> * Vision models (e.g., CNNs, ResNets)
> * Transformers (we are currently building Transformer+XNet models)
>   * On **small-scale Transformers**, we observe particularly strong performance, especially when replacing the softmax in attention mechanisms with **heavy-tailed Cauchy kernels**
>   * On **large-scale Transformers**, the effects are currently **less pronounced**, and we are conducting further research to understand this phenomenon
> * Recurrent or residual blocks
>
> We expect XNet to generalize well in perception/language domains, and early results in machine translation are encouraging.
>
> ### (Q2) Deep XNets and training speed vs. model size
>
> In practice, deep XNets **train faster to reach a given accuracy** than ReLU- or GELU-based counterparts, especially on smooth or structured data (see results in [1]).
>
> While the Cauchy activation has slightly more compute per unit (due to the element-wise division), this is **negligible in total runtime**, and is **outweighed by faster convergence and better performance-per-FLOP**.
>
> We will include **runtime plots and scaling tables** in the appendix.
>
> ### (Q3) Parallelization bottlenecks?
>
> No. Cauchy activations are **element-wise and differentiable**, and thus parallelize just as efficiently as ReLU, GELU, etc. There are **no reductions or cross-sample dependencies** that hinder GPU or TPU execution.
>
> That said, in preliminary experiments on **very large-scale models (e.g., 10B+ parameters)**, we observed some **degradation in performance with XNet**. We are currently investigating this phenomenon and believe it may be due to interaction between large activations and normalization layers. We plan to report detailed findings in future work.
>
> ---
> ### References
> [1] Li, X., Zhang, H, and Xia, Z. "Cauchy activation functions and XNet." *Neural Networks* 188, 107375 (2025).

---

> > ### Author Response · Authors · 2025-08-07
> >
> > Dear Reviewer,
> >
> > Thank you again for your constructive and encouraging review.
> >
> > As the rebuttal phase is approaching its end, we would greatly appreciate it if you could kindly take a moment to check our response and the additional experiments we have included.
> >
> > In particular, we have provided detailed theoretical clarifications and new empirical results addressing your valuable comments.
> >
> > Your confirmation or further suggestions would be very helpful in improving our work.
> >
> >
> > Warm regards,

---

### Public Comment · ~Jaemin_Oh1 · 2025-11-05
**reference [3]**

Dear the authors, the *rational neural networks* paper was published in NeurIPS 2020, not in JCP 2021.

---

> ### Public Comment · ~Xin_Li73 · 2025-12-11
>
> Thanks for the reminder!

---

### Decision · Program_Chairs · 2025-09-17

**Decision:**

Accept (poster)

**Comment:**

The reviewers agreed that this paper addresses an important topic, is clearly written, and has strong results. Some reviewers had concerns about the variety of tasks, models, and datasets in the experimental validations. The authors' rebuttal and further answers during the discussion period seems satisfactory. Given the unanimous recommendation for acceptance from all the four reviewers, I recommend accepting this paper.